# Regulatory protein SrpA controls phage infection and core cellular processes in *Pseudomonas aeruginosa*

Jiajia You[1], Li Sun[1], Xiaojing Yang[1], Xuewei Pan[1], Zhiwei Huang[1], Xixi Zhang[1], Mengxin Gong[1], Zheng Fan[2], Lingyan Li[1], Xiaoli Cui[1], Zhaoyuan Jing[1], Shouguang Jin[2,4], Zhiming Rao[3], Weihui Wu[2] & Hongjiang Yang[1]

Our understanding of the molecular mechanisms behind bacteria-phage interactions remains limited. Here we report that a small protein, SrpA, controls core cellular processes in response to phage infection and environmental signals in *Pseudomonas aeruginosa*. We show that SrpA is essential for efficient genome replication of phage K5, and controls transcription by binding to a palindromic sequence upstream of the phage RNA polymerase gene. We identify potential SrpA-binding sites in 66 promoter regions across the *P. aeruginosa* genome, and experimentally validate direct binding of SrpA to some of these sites. Using transcriptomics and further experiments, we show that SrpA, directly or indirectly, regulates many cellular processes including cell motility, chemotaxis, biofilm formation, pyocyanin synthesis and protein secretion, as well as virulence in a *Caenorhabditis elegans* model of infection. Further research on SrpA and similar proteins, which are widely present in many other bacteria, is warranted.

[1] State Key Laboratory of Food Nutrition and Safety, Key Laboratory of Industrial Microbiology of the Ministry of Education, Tianjin Key Laboratory of Industrial Microbiology, College of Biotechnology, Tianjin University of Science and Technology, Tianjin 300457, China. [2] Department of Microbiology, State Key Laboratory of Medicinal Chemical Biology, Key Laboratory of Molecular Microbiology and Technology of the Ministry of Education, College of Life Sciences, Nankai University, Tianjin 300071, China. [3] The Key Laboratory of Industrial Biotechnology, Ministry of Education, Laboratory of Applied Microorganisms and Metabolic Engineering, School of Biotechnology, Jiangnan University, Wuxi 214122, China. [4] Department of Molecular Genetics and Microbiology, College of Medicine, University of Florida, Gainesville, FL 32610, USA. Correspondence and requests for materials should be addressed to Z.R. (email: raozhm@jiangnan.edu.cn) or to W.W. (email: wuweihui@nankai.edu.cn) or to H.Y. (email: hongjiangyang@tust.edu.cn)

Phage and bacteria are the two most abundant biological entities co-existing in almost every type of natural ecosystems, accompanied by constant predation and anti-predation events between them[1]. During a long history of reciprocal competition with phages, bacteria have evolved several genetic systems that actively stop phage attacks, such as restriction-modification (R-M) system, CRISPR-Cas system, abortive infection (Abi) system, toxin-antitoxin (TA) system, and bacteriophage exclusion (BREX) system[2, 3]. Besides these anti-phage weapons, some metabolic pathways help bacteria to avoid or reduce phage infection. Quorum sensing (QS) system plays key roles in multiple core cellular pathways, and as a side effect, it can render host cells resistant to phage infections by reducing the amount of phage receptors on cell surface[4]. Also, the QS system can affect cell physiological state and cell populations, leading to attenuated phage reproduction[5]. Extracellular polymers, such as exopolysaccharides, lipopolysaccharide (LPS), capsule, and proteinaceous layer, may function as physical shields against phage adsorption by masking phage receptors[6, 7]. These bacterial genes are employed to defend against phage invasions actively as well as passively.

On the other hand, bacterial genes are also required for successful phage infection. Individual phage-resistant mutants are often selected from sensitive bacterial populations with diverse genetic mutations. Most mutants show decreased adsorption rate with mutations occurring in the genes involved in receptor synthesis, such as LPS, flagella, pili, fimbriae, outer membrane proteins, and other cell appendages. A small number of mutants show wild-type adsorption rate (WAR) with genetic mutations occurring in metabolic pathways that are essential for various phage infection stages, such as genome injection, transcription, replication, and packaging[7–9].

Although a number of bacterial genes have been shown essential for phage reproduction, our knowledge on their functions is still limited, considering the enormous genetic diversity of both bacteria and phages.

Phage K5 is a lytic myovirus specific to *Pseudomonas aeruginosa* strain PAK. Previously, a Tn5G transposon library of *P. aeruginosa* PAK-AR2 was constructed and screened for bacterial genes required for the phage K5 infection[10].

Here, we select a WAR type mutant 1–9 and show that the Tn5G transposon is inserted in a gene encoding a hypothetical small regulatory protein (SrpA) of 84 amino acids long, belonging to the HTH_XRE super family. We investigate the regulatory role of the SrpA protein in the phage K5 infection and the diverse cellular metabolic pathways. Our data show that the small regulator SrpA plays versatile regulatory functions by binding directly to the promoter region of the phage RNA polymerase gene as well as that of several bacterial genes, influencing diverse cellular processes that include type III secretion system, chemotaxis, cell motility, cell shape, type VI secretion system, pyocyanin synthesis, and biofilm formation.

## Results

**Identification of a regulator that controls phage infection.** A Tn5G transposon insertion library of PAK-AR2 had previously been screened for phage K5-resistant mutants[10]. Among the phage-resistant mutants, strain 1–9 displayed a WAR as its parent strain PAK-AR2 (Fig. 1a). Inverse PCR and sequencing identified the transposon Tn5G being inserted right behind 37th bp of a gene Y880_RS18270 in the updated draft genome of *P. aeruginosa* PAK-Assembly GCF_000568855.1: scaffold00004 (NZ_KK037228.1). Further bioinformatics analysis showed that the disrupted gene encodes a hypothetical small regulatory protein (SrpA) with a helix-turn-helix (HTH) DNA-binding domain that belongs to the XRE super family (Fig. 1b). In the complementation assay, introduction of the *srpA* gene (pLLY1101) into the strain 1–9 restored its sensitivity to the phage K5 (Fig. 1c). These results indicated that the bacterial *srpA* gene is required for a successful infection by the phage K5.

**SrpA positively regulates virus replication.** Although strain 1–9 shows WAR phenotype and resistant to the phage K5, growth curve analysis indicated that infection by the phage K5 significantly reduces the growth rate of strain 1–9 (Fig. 2a). These results suggested that the viral genome was possibly injected into the mutant cell and initiated a series of viral activities[8].

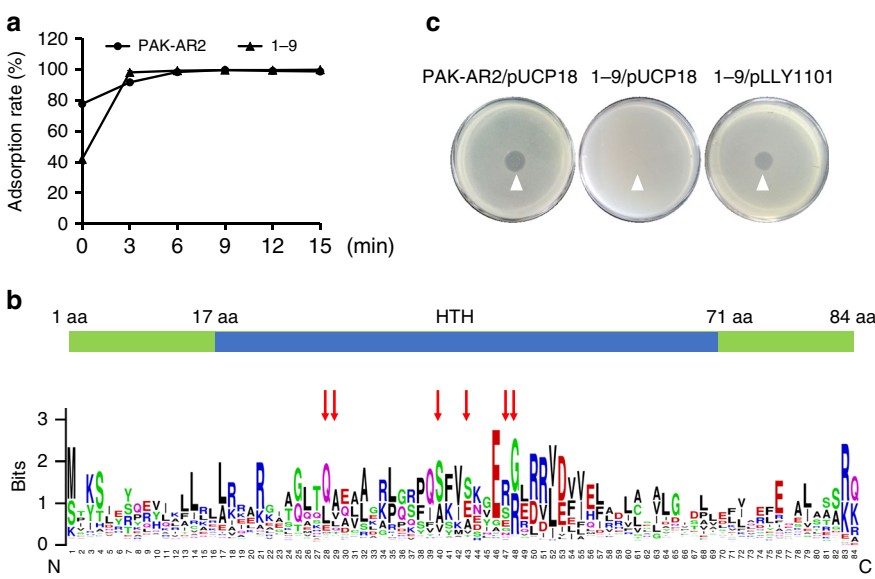

**Fig. 1** Characterization of the *srpA* mutant. **a** Time course of the phage adsorption rate. PAK-AR2: sensitive to the phage K5. 1–9: resistant to the phage K5. **b** Protein structure of the SrpA. Red arrows represent the amino acids recognizing specific DNA sequences. Larger letters represent higher conservative amino acid residues. **c** Complement test by spotting assay. Plasmid pLLY1101 carries a *srpA* gene. The experiment was independently replicated three times (**a**). Error bars show standard deviations. HTH helix-turn-helix DNA-binding domain

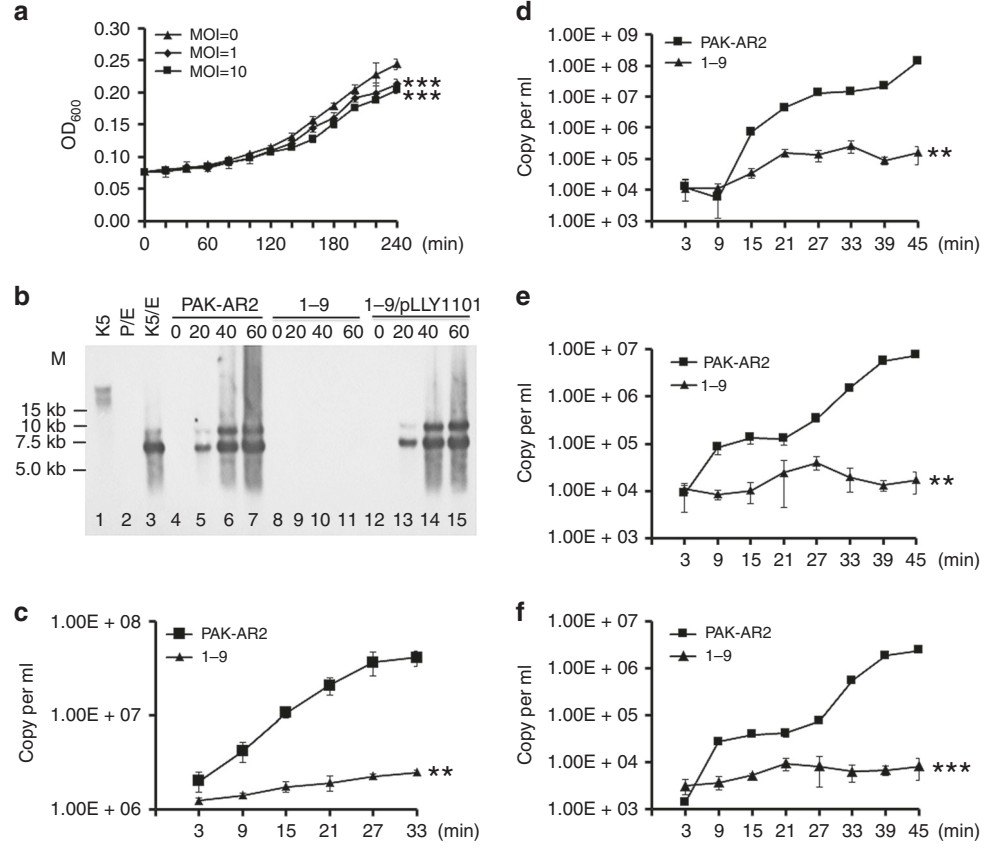

**Fig. 2** SrpA is a positive regulator for phage K5 replication and transcription. **a** Growth inhibition curves of strain 1–9 by the phage K5. **b** Southern blot analysis. Total DNA was isolated at the indicated time intervals, digested with *Eco*RV, and then hybridized with a phage-derived DNA probe (see Methods for details). Plasmid pLLY1101 carries the *srpA* gene. Lane 1 (K5): phage K5 genomic DNA. Lane 2 (P/E): PAK-AR2 genomic DNA digested with *Eco*RV. Lane 3 (K5/E): genomic DNA of the phage K5 digested with *Eco*RV. Lane 4-7: total DNA samples from the PAK-AR2 cells infected with the phage K5. Lane 8-11: total DNA samples from the 1–9 cells infected with the phage K5. Lane 12-15: total DNA samples from the 1–9/pLLY1101 cells infected with the phage K5. **c** RT-qPCR analysis of the phage genome in the indicated strains. **d–f** Target gene expression analysis by RT-PCR. Expression of the phage RNA polymerase gene (**d**), the phage DNA polymerase I gene (**e**), and the phage DNA polymerase II gene (**f**). The experiments were independently replicated three times (**a**, **c–f**). One-way ANOVA was used to examine the mean differences between the end points of the data groups. \*\*$P < 0.01$. \*\*\*$P < 0.001$. Error bars show standard deviations. MOI multiplicity of infection

Southern hybridization was conducted to analyze the genome replication efficiency and packaging process of the phage K5. Both PAK-AR2 and 1–9/pLLY1101 displayed similar positive profiles, whereas no positive bands were detected in strain 1–9 (Fig. 2b), indicating that SrpA is required for the phage K5 genome replication. In the Southern blot analysis, two positive *Eco*RV bands were detected, a 6002 bp fragment from the 5′ end of the monomeric genomic DNA, and an 8724 bp fragment from the end junction of the concatemeric genomic DNA (Fig. 2b). The data indicated that the phage K5 packages its genome in a way similar to that of the phage C11 as described previously[8].

Real-time quantitative PCR was further conducted to analyze the replication efficiency of the phage K5 in strain 1–9 and PAK-AR2. After 30 min infection, a 2.0-fold increase of the K5 genome copy numbers was observed in strain 1–9, while 19.5-fold increase was observed in PAK-AR2 (Fig. 2c). These data demonstrated that the viral genome was successfully injected into the bacterial cells and genome replication process was initiated, although at a much lower replication rate in the strain 1–9, only 6% that of the parent strain PAK-AR2, possibly explaining the phage-resistant phenotype of the strain 1–9 (Fig. 2c).

Expression of the phage genes for RNA polymerase, phage DNA polymerase I gene, and phage DNA polymerase II was

tested by Quantitative real-time RT-PCR method. Compared to their expression in strain PAK-AR2, all three genes showed significantly decreased levels of expression in the mutant strain 1–9 (Figs. 2d–f). These data indicated that SrpA was required for efficient transcription of these key phage genes.

**SrpA and phage RNA polymerase co-regulate virus transcription.** With a transcriptional *lacZ* reporter fusion, expression of the RNA polymerase gene *gp058* was found significantly higher in PAK-AR2 when infected with the phage K5, while no difference was observed in the strain 1–9 (Fig. 3a). These results suggested that unknown viral factor(s) from the phage K5 might enhance the transcription of the *gp058* gene in PAK-AR2 cells, but not in the 1–9 cells. Similarly, the expression of additional phage genes was further assessed using the transcriptional *lacZ* reporter fusions, including the DNA polymerase I gene *gp104*, the DNA polymerase II gene *gp105*, the major capsid protein gene *gp056*, the base plate protein gene *gp071*, and the ribonucleotide diphosphate reductase β-subunit gene *gp128*. The assay results showed that transcription of all these genes is dependent on the presence of the regulator SrpA (Fig. 3b, c). Moreover, the expression levels of these phage genes in PAK-AR2 showed significant increases when *gp058* was overexpressed, by 44.7%

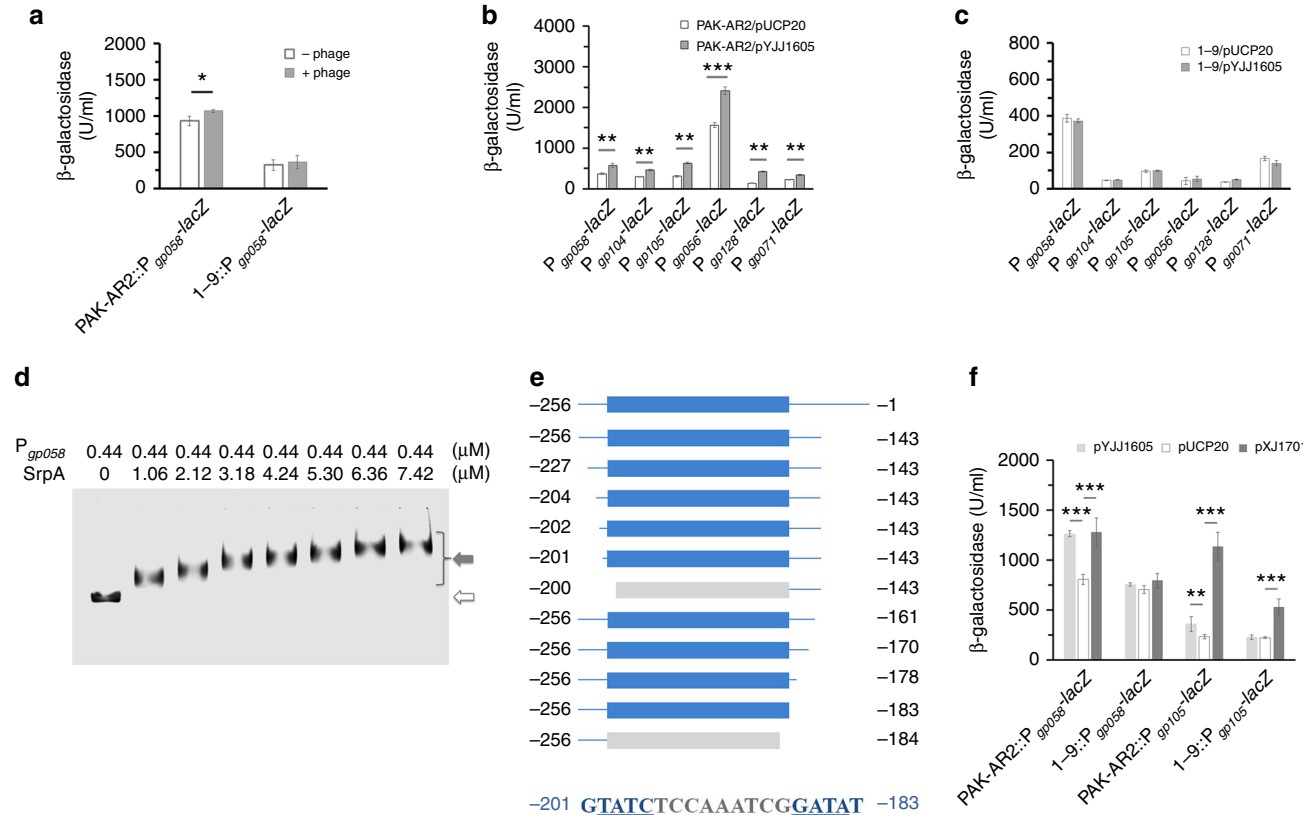

**Fig. 3** Regulatory function of the SrpA. **a** Phage K5 infection improves the transcription of the phage RNA polymerase gene *gp058* in PAK-AR2. **b** Transcription profiles of the indicated phage genes in PAK-AR2 carrying pYJJ1605, which expresses the phage RNA polymerase gene *gp058* driven by its native promoter. **c** Transcription profiles of the indicated phage genes in 1–9 carrying pYJJ1605. **d** EMSA assay for the SrpA binding of the *gp058* promoter. The gray arrow indicates bound DNA fragments. The blank arrow indicates free DNA fragments. **e** Mapping of the SrpA-binding sequence by EMSA assay. Fragments with the intact binding site (blue color) showed positive bindings. Fragments with the defective binding site (gray color) showed no bindings. Arabic numbers represent the distances (bp) upstream from the start codon ATG of the *srpA* gene. **f** β-galactosidase activity assay of strains PAK-AR2 and 1–9 harboring either the $P_{gp058}$-*lacZ* (with a SrpA-binding site) or $P_{gp105}$-*lacZ* (without a SrpA-binding site) reporter fusion. The plasmid pYJJ1605 and pXJ1701 express the phage RNA polymerase gene *gp058* driven by its native promoter and the $P_{lac}$ promoter, respectively. The experiments were independently replicated three times and one-way ANOVA was used to examine the mean differences between the data groups (**a–c**, **f**). *$P < 0.05$. **$P < 0.01$. ***$P < 0.001$. Error bars show standard deviations

($P < 0.01$, one-way analysis of variance (ANOVA)), 58.2% ($P < 0.01$), 106.1% ($P < 0.01$), 54.6% ($P < 0.001$), 54.5% ($P < 0.01$), and 217.7% ($P < 0.05$), respectively, indicating that the RNA polymerase encoded by the phage K5 was necessary for efficient transcription of those phage genes (Fig. 3b). However, almost no effect was observed when strain 1–9 was used as the host (Fig. 3c). These results indicated that SrpA and the phage RNA polymerase possibly co-regulate transcription of phage genes. Bacterial two-hybrid assay was further conducted and a positive interaction was indeed observed between these two proteins (Supplementary Fig. 1).

To further explore the regulatory mechanism, purified His-tagged SrpA protein was incubated with a fluorescence-labeled promoter fragment of the *gp058* gene and subjected to EMSA experiment. The gel mobility shift patterns showed that SrpA specifically bound to the *gp058* gene promoter (Fig. 3d). A series of PCR products were further tested in the EMSA assay to map the binding site, and the sequence located between −201 and −183 bp upstream of the ATG start codon was identified as the target (5′-GTATCTCCAAATCGGATAT-3′), which contains a palindromic structure (Fig. 3e). Searching the K5 genome, a similar binding site (5′-CTATCTGCCTCCTCGATAT-3′) was found in the promoter of the N6 adenine-specific DNA methyltransferase gene (*gp053*), and another (5′-TTATC-GAGGCGTTGGATAT-3′) in the promoter of a hypothetical protein gene (*gp125*).

To further verify the co-regulatory partnership between SrpA and Gp058, a transcriptional fusion of $P_{gp105}$-*lacZ* was constructed where the *gp105* promoter lacks a SrpA-binding site. In the presence of a $P_{lac}$ promoter driven phage RNA polymerase gene (carried on the plasmid pXJ1701), high level of $P_{gp105}$-*lacZ* expression was observed in both PAK-AR2 and 1–9. However, when the phage RNA polymerase gene was driven by its native promoter (carried on the plasmid pYJJ1605), no significant expression was observed in the strain 1–9, compared to that in PAK-AR2. These data suggested that the transcription of $P_{gp105}$-*lacZ* is independent of the SrpA, while that of $P_{gp058}$-*lacZ* requires both SrpA and Gp058 (Fig. 3f).

**Transcriptome analysis of the *srpA* mutant strain.** More and more small proteins that were ignored from typical genome annotations have now been experimentally demonstrated to play important regulatory roles on various bacterial metabolic pathways[11, 12]. As a small hypothetical regulator encoded by bacteria, SrpA may also play regulatory role in the cellular metabolism. RNAseq analysis was performed to evaluate gene expression differences between strains 1–9 and PAK-AR2. Transcriptome data showed that expression of 361 genes were upregulated while 459 genes were downregulated by at least twofold when comparing the *srpA* mutant strain 1–9 to its parent

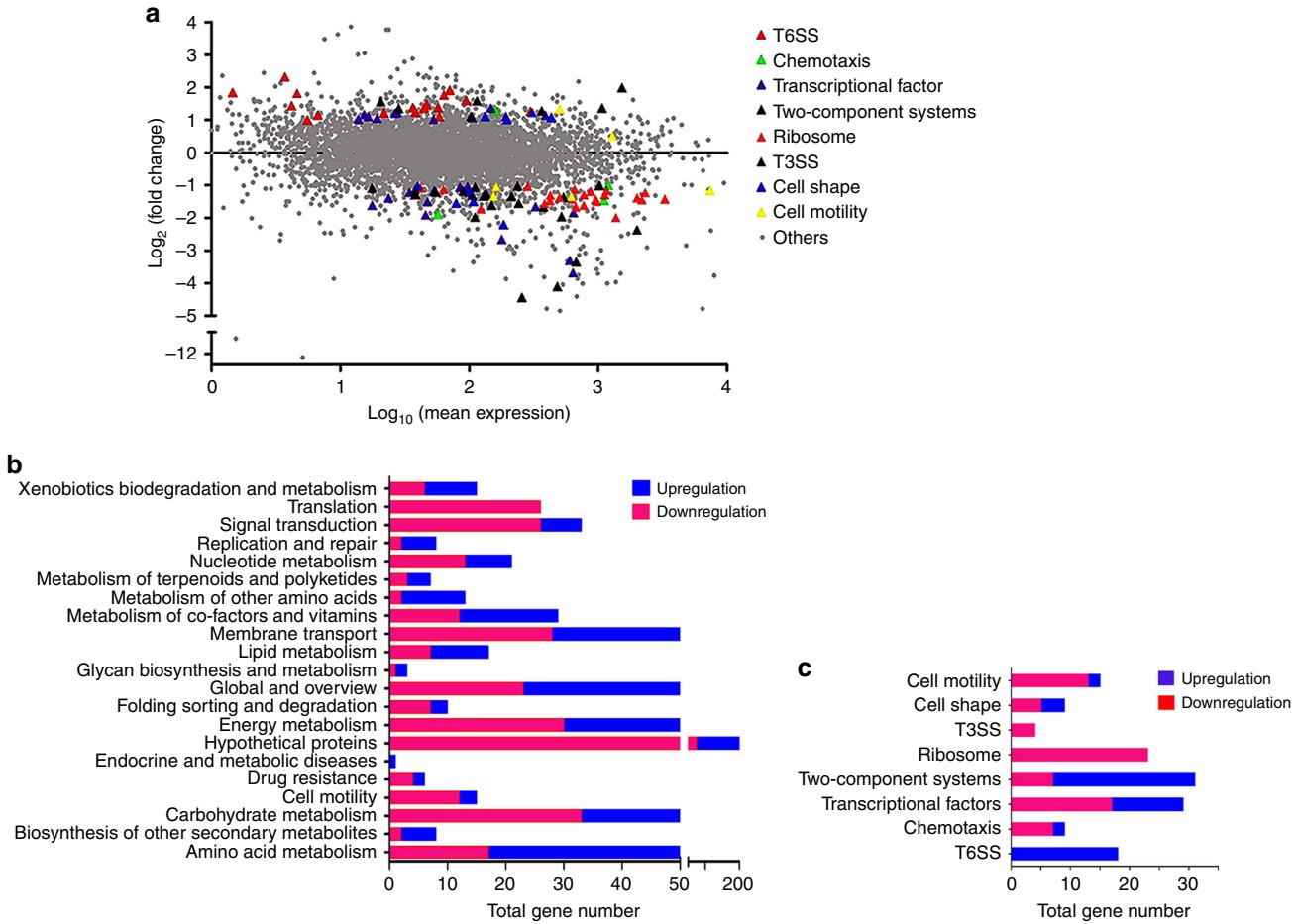

**Fig. 4** Transcriptome analysis of strains 1–9 and PAK-AR2. **a** X-axis represents the logarithm-transformed value of gene expression levels. Y-axis represents log2-transformed value of expression change folds. Genes belonging to different pathways are represented by various colored shapes as indicated. Others represent genes not belonging to the indicated pathways. **b** Expression profiles of the genes belonging to the cellular processes in KEGG_B_class. **c** Expression profiles of the genes belonging to the indicated metabolic pathways. Red color shows the genes with significantly reduced transcription levels. Blue color shows the genes with significantly elevated transcription levels

strain PAK-AR2 (Fig. 4a). These genes were classified into 21 major cellular processes based on the annotation of KEGG_B_-class (Fig. 4b) or further grouped into several major metabolic pathways, such as ribosomal proteins, type III secretion system (T3SS), type VI secretion system (T6SS), chemotaxis, cell motility, and cell shape control (Fig. 4c). These data suggested that SrpA possibly functions as a key regulator controlling versatile cellular functions.

**SrpA regulates cell motility and biofilm formation**. Swimming, swarming, and twitching are three modes of motilities found in *P. aeruginosa*. When tested for swimming motility with semi-solid agarose plates, no significant difference was observed among PAK-AR2, 1–9, and 1–9/pLLY1101, suggesting SrpA has little effect on bacterial swimming motility (Fig. 5a, upper panel). For swarming test, absence of SrpA (strain 1–9) resulted in a typical dendritic structure, which was not observed in PAK-AR2 or 1–9/pLLY1101, indicating that SrpA represses swarming motility under the tested condition (Fig. 5a, middle panel). In twitching motility tests, strains PAK-AR2 and 1–9/pLLY1101 showed high twitching activities with the diameters of twitching zones about 2.5-fold ($P < 0.05$, one-way ANOVA) that of strain 1–9 (Fig. 5a, lower panel and 5b), suggesting that SrpA is required for the twitching motility.

Chemotaxis activity was further assessed using casein hydrolysates as the inducing reagent. Cells of strain 1–9 moved more rapidly and efficiently, as the migrating bacterial cell numbers were 3.6-fold and 3.1-fold that of PAK-AR2 and 1–9/pLLY1101, respectively (Fig. 5c). Transcription of 20 chemotaxis-related genes was found significantly changed in the *srpA* mutant, with elevated expression of *ercS"*, *flgB*, *gltR*, *PA1646*, *phoU*, and *pilE* (Supplementary Data 1). *PA1646* encodes a putative methyl-accepting chemotaxis protein functioning as a probable chemotaxis transducer. The increased *PA1646* transcription may partially explain an elevated chemotaxis activity of the strain 1–9[13].

Biofilm is also an important virulence factor of *P. aeruginosa* and its formation is regulated by multiple pathways[14]. In the *srpA* mutant, the transcription of biofilm-related regulators was dramatically downregulated, including *rhlI*, *fis*, *suhB*, and *tspR*, while others were upregulated, such as *rsmA* that negatively regulates the biofilm formation (Supplementary Data 1). The effect of SrpA on biofilm formation was evaluated, and the results indeed showed that the strain 1–9 produced much less biofilm than PAK-AR2 or 1–9/pLLY1101 (Fig. 5d), indicating that SrpA also positively influence the biofilm formation.

**SrpA is required for a full bacterial virulence**. Pyocyanin is an important virulence factor of *P. aeruginosa* and can induce

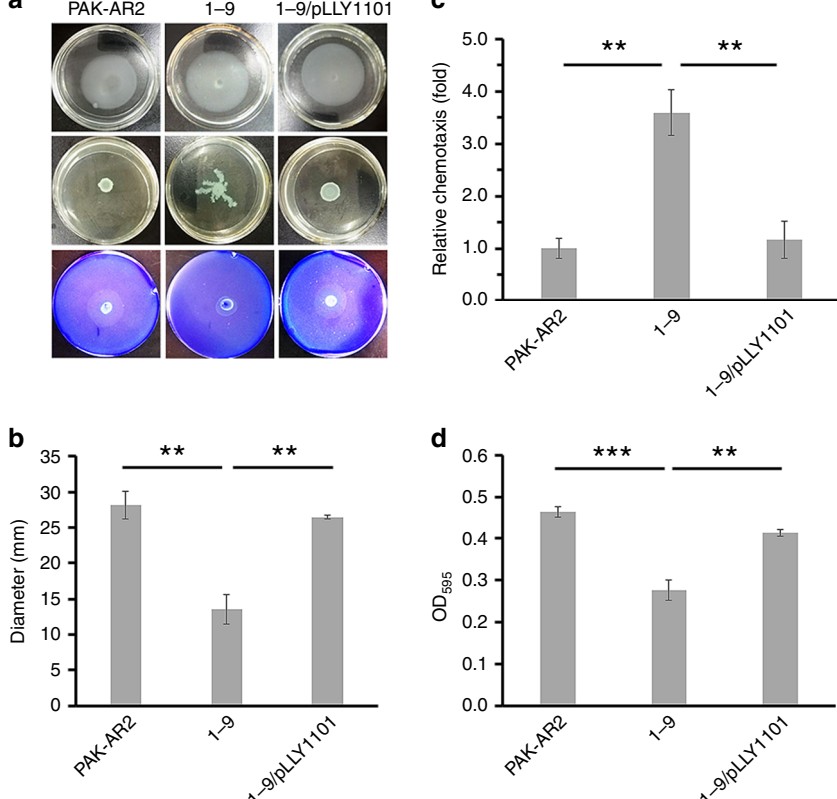

**Fig. 5** Assays for motility, chemotaxis, and biofilm formation. **a** Motility tests for PAK-AR2 (left), 1–9 (middle), and 1–9/pLLY1101 (right). Upper panel: swimming assay. Middle panel: swarming assay. Lower panel and (**b**) twitching motility assay. **c** Chemotaxis activity evaluation of cell response to the induction by casein hydrolysates. **d** Biofilm formation assay. pLLY1101 carries the *srpA* gene. Triplicates of each samples were tested. The experiments were independently replicated three times. One-way ANOVA was used to examine the mean differences between the data groups. **P < 0.01. ***P < 0.001. Error bars show standard deviations

neutrophil apoptosis and necrosis in cystic fibrosis (CF) infection[15]. The pyocyanin synthesis was investigated in the *srpA* mutant. No difference in the pyocyanin production was observed between PAK-AR2 and 1–9, while strain C11-1, a *srpA* mutant of PAK produced less amount of pyocyanin than PAK ($P < 0.1$, one-way ANOVA) and C11-1/pLLY1101 ($P < 0.01$), respectively (Fig. 6a). These results suggested that the *purEK* defect might affect pyocyanin synthesis in the strain PAK-AR2 and thus masks the SrpA-mediated regulatory effect.

T3SS of *P. aeruginosa* is a dominant virulence factor in acute infections. The expression of T3SS is directly controlled by ExsA, ExsC, ExsD, and ExsE, which form a regulatory cascade that specifically responds to the environmental signals[16]. Transcriptome analysis showed that the transcription of *exsA*, encoding a master activator for the T3SS, was decreased significantly in the strain 1–9 (Supplementary Data 1). This data was further verified by western blot analysis, with the strain 1–9 secreting much less amount of ExoS and ExoT than that of PAK-AR2 and 1–9/pLLY1101 under T3SS-inducing condition (Fig. 6b), suggesting that SrpA positively regulates T3SS expression.

To further explore the in vivo role of SrpA, worm killing assay was conducted. Two sets of strains were tested, two parental strains PAK and PAK-AR2, with their corresponding *srpA* gene mutants, and the two *srpA* mutants complemented by intact *srpA* gene. The assay results showed that disruption of the *srpA* gene compromised the bacterial virulence significantly in both groups (Fig. 6c). These data indicated that SrpA is required for a full virulence of *P. aeruginosa* in *C. elegans* infection model.

**SrpA inhibits T6SS-mediated antibacterial activity.** *P. aeruginosa* encodes three sets of T6SS, H1-T6SS, H2-T6SS, and H3-T6SS, and uses them as weapons against prokaryotic or eukaryotic target cells with diverse effectors[17, 18]. Transcriptome data highlighted that a number of the T6SS-related genes were upregulated significantly in the *srpA* mutant (Supplementary Data 1). To verify the data further, *Pseudomonas* strains were co-cultured with *Escherichia coli* DH5α carrying a *gfp* gene to test their antibacterial activities. After 120 min of co-incubation of the strains, 39.3% of *E. coli* cells was killed in the group with strain 1–9, while only 0.9% of *E. coli* cells was killed in the group with the strain PAK-AR2 (Fig. 7a, b), indicating that SrpA inhibits the T6SS-mediated antibacterial activity of the *P. aeruginosa* strains.

**SrpA is required to maintain bacterial cell shape.** MreB, MreC, and MreD are actin-like proteins encoded by the genes in the PA4479-PA4480-PA4481 operon. MreB is essential for intracellular protein transportation and plays key roles in cell wall synthesis, cell shape maintenance, cell growth, chromosome segregation, and polar localization of motility-related proteins[19–21]. Transcriptome data revealed that the expression of these three genes was dramatically reduced in the *srpA* mutant (Supplementary Data 1). Strain 1–9 exhibited much smaller sized colonies on L-agar plates. Conventional optical microscopy showed that the 1–9 cells are dot-shaped, significantly shorter but wider than the PAK-AR2 cells (Fig. 7c, upper panel). To quantitatively characterize the cell shape of strain 1–9, cell images were taken using SEM. The average length of the 1–9 cells was 31.1% shorter than

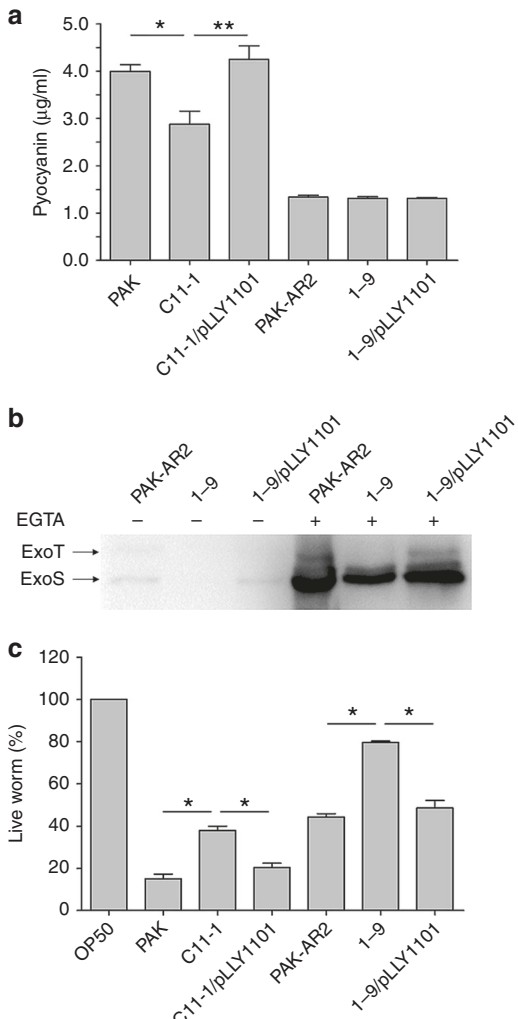

**Fig. 6** Virulence assessment of the strain 1–9. **a** Analysis of pyocyanin produced by the indicated *P. aeruginosa* strains. **b** Detection of secreted ExoS and ExoT in bacterial culture supernatants by western blot. Full blots are shown in Supplementary Fig. 6. **c** Worm killing assay. OP50: *E. coli* strain used for feeding worms. C11-1: a *srpA* mutant derivative of PAK. 1–9: a *srpA* derivative of PAK-AR2. pLLY1101: a plasmid carrying the *srpA* gene. Triplicates of each samples were tested. The experiments were independently replicated three times. One-way ANOVA was used to examine the mean differences between the data groups. *$P < 0.05$. **$P < 0.01$. Error bars show standard deviations

that of the PAK-AR2 cells, while the average width of the 1–9 cells was 23.1% wider than that of the PAK-AR2 cells (Fig. 7c, lower panel and 7d). These data indicated that SrpA is required to maintain bacterial cell shape, possibly through positive regulation of the *mreB*, *mreC*, and *mreD* genes.

**Identification of SrpA-binding sites in bacterial genome**. SrpA can bind to a palindromic sequence in the promoter of the *gp058* gene. This sequence and its derivative sequences (5′-NTATCN (0,9)GATAN-3′) were further searched across the draft genome sequence of strain PAK (http://www.pseudomonas.com/replicon/setmotif) and 375 potential SrpA-binding sites were identified. Among them, 66 were found in the intergenic regions proximal to the promoters of various genes, including 6 regulators with known functions (AntR, QscR, TspR, CzcR, RtcR, and PhhR), 9 regulators with unknown function, and 21 genes encoding metabolic enzymes and structural proteins, such as FlgB, FimV,

MucR, RpsG (S7), RpsL (S12), and RpsP (S16) (Fig. 8). The remaining 30 genes encode putative proteins and hypothetical proteins (Supplementary Table 1). To verify the SrpA-binding sites, selected promoter regions were amplified and the PCR products were subjected to EMSA test, including three bacterial genes *PA1826*, *flgB*, *PA3766*, and three phage genes, *gp053*, *gp104*, and *gp125*. The *gp105* promoter that lacks a SrpA-binding site was used as a negative control. The results showed that the sequences that contain the palindromic structure can all be bound by the SrpA protein with varying affinities (Supplementary Fig. 2). Additionally, the SrpA-binding sequences on the test promoters may associate with varying number of SrpA and form different secondary structures, contributing to the various DNA migration patterns (Fig. 3d and Supplementary Fig. 2). More experiments are required to clarify the factors influencing the interaction between SrpA and its diverse binding sequences.

**SrpA-like proteins are widely distributed among bacteria**. Homologs of SrpA were searched with an *E*-value lower than 5*E*−08. The top 1000 proteins show similarities ranging 55.3–100% and are widely found in the groups of γ-proteobacteria, enterobacteria, α-proteobacteria, high GC Gram-positive bacteria, and β-proteobacteria (Supplementary Data 2 and Fig. 9a). One homolog was found in archaeon GW2011_AR11 (groundwater metagenome) with a relatively high similarity (67.2%). With *E*-value set lower than 0.001, 120 SrpA-like *P. aeruginosa* proteins and 25 SrpA-like phage proteins were identified in the nonredundant protein sequences (nr) database. When the genome database of *Pseudomonas* was searched, 210 SrpA-like proteins with *E*-value lower than 9.2*E*−17 were identified and 7 of them are also present in the dataset of 120 SrpA homologs. In the phylogenetic analysis, three groups of SrpA-like proteins clustered in three major clades. Interestingly, one bacterial SrpA-like protein (WP_034058829.1) is positioned within the phage clade (Supplementary Data 3, Supplementary Data 4, and Fig. 9b). The fact that the highly homologous SrpA is rarely distributed in both archaea and phages suggests that SrpA is likely originated from the bacterial ancestor.

The phylogenetic tree of 2156 strains with complete or incomplete draft genomes was created on the basis of the available sequences of the 16s ribosomal RNA (rRNA) genes. The *srpA* homologs were widely present in about 9.47% strains and clustered in several major clades. However, *srpA* homologs were not found in the reference strains of PAO1 and PA14 (Supplementary Data 5 and Fig. 9c). Interestingly, the *srpA* gene from PAK shows similar effects on the cell motility when introduced into the strains PAO1 or PA14 (Supplementary Fig. 3). On the PAK genome, the *srpA* gene is located between two coding genes, the DNA polymerase subunits gamma/tau (*Y880_RS18245*, *danX*) and the DNA ligase (*Y880_RS18290*, *lig*). The homologs of these two genes are found in PAO1 and PA14 (Supplementary Fig. 4). In comparison, the genome of PAK has an extra 7.2 kb fragment inserted between the genes *danX* and *lig*, which includes the *srpA* gene, several hypothetical protein genes, and a putative integrase gene. The genomic context indicates that this fragment was probably acquired via horizontal gene transfer, which may even depend on the integrase.

## Discussion

Besides the well-studied bacterial genes associated with phage receptor synthesis, a few other host genes that play important roles in the phage infection process have also been thoroughly investigated, such as the *E. coli* RNA polymerase required for the early genes transcription of T7 phage[22], the *E. coli* thioredoxin required for the stabilization of the T7 replisome complex by

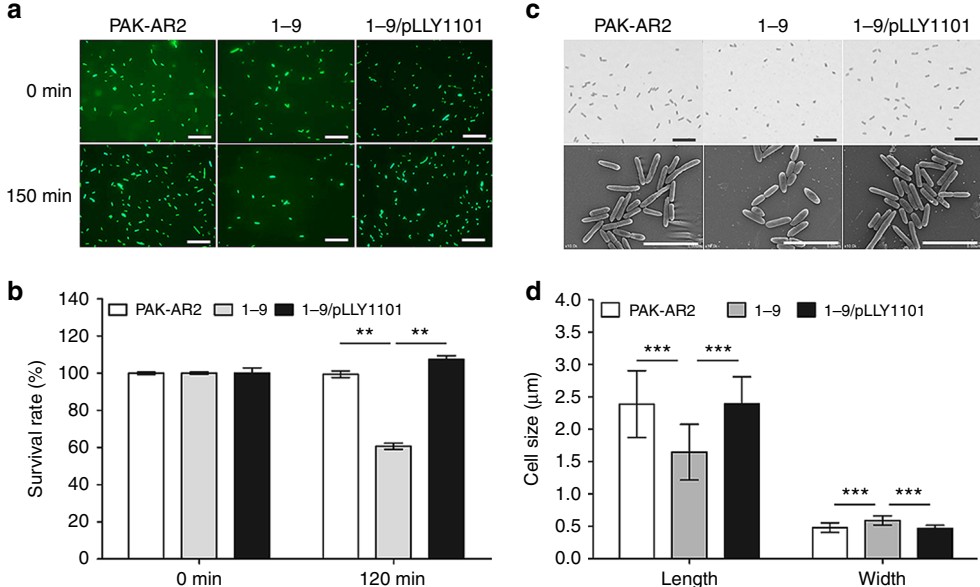

**Fig. 7** Influence of SrpA on T6SS activity and cell shape. **a** Fluorescent microscopy images of *E. coli* DH5α carrying the *gfp* gene incubated with the indicated *P. aeruginosa* strains. The scale bars represent 10 μm. **b** Enumeration of viable *E. coli* cells using flow cytometry (lower panel). **c** Gram-stained bacterial cells of the indicated strains and the scale bars represent 10 μm (upper panel). SEM images of the indicated cells and the scale bars represent 5 μm (lower panel). **d** Measurements of cell dimensions (40 cells per strain, lower panel). The experiments were independently replicated three times. One-way ANOVA was used to examine the mean differences between the data groups. $**P < 0.01$. $***P < 0.001$. Error bars show standard deviations

direct interaction with T7 DNA polymerase[23], and the *B. subtilis* DNA polymerase genes *polC* and *dnaE* required for SPP1 DNA replication[24]. Moreover, many bacterial genes were found associated with phage infection process by genome-wide screens without further details, such as *E. coli* genes from the ASKA library and the Keio collection for λ or T7 phage infection[9, 25], *B. subtilis* genes for SPP1 phage infection[26], and *P. aeruginosa* genes from a Tn5G insertion bank for C11 phage infection[8]. Recently, global transcription changes of bacterial genes were investigated following infection by phages, such as genetic response of *P. aeruginosa* to PaP3 infection[27], *Lactococcus lactis* subsp. *lactis* IL1403 to the lytic phage c2 infection[28], *B. subtilis* to the giant phage AR9 infection[29], *P. aeruginosa* to the phage PAK_P3 and PAK_P4 infections[30], and *B. subtilis* to phage ϕ29 infection[31]. However, the transcriptome profiling provided limited information on the requirement of bacterial genes for phage infection. In this study, a small hypothetical protein SrpA was confirmed to function as a regulator by directly binding to the promoter region of the phage RNA polymerase gene for transcription initiation. Moreover, SrpA shows specific interaction with the phage RNA polymerase inferred from the bacterial two-hybrid experimental results. SrpA may control phage gene transcription through recruiting the phage RNA polymerase during infection process, e.g., the *gp053* gene encoding a N6 adenine-specific DNA methyltransferase has a SrpA-binding site at its promoter region. The methyltransferase can be transcribed simultaneously with the phage RNA polymerase gene and it may protect the phage genome from digestion by the host restriction endonucleases through DNA methylation[32]. Identical amino acid sequence of the RNA polymerases has been found in many more *P. aeruginosa* phages, such as K8, C11, JG004, PAK_P1, PAK_P2, phiMK, PA10, and vB_PaeM_C2-10_Ab02, with the SrpA-binding sequences present in the promoter regions of all these RNA polymerase genes. Although these *Pseudomonas* phages were recovered from different natural sites across the world[8], they might share the same infection mode as that of the phage K5.

*P. aeruginosa* is an opportunistic human pathogen that can cause severe infections in human beings especially in patients

with compromised immunity. Virulence factors involved in *P. aeruginosa* infections have been well documented. T3SS and type IV pili are the major determinants critical for acute infections, whereas T6SS, biofilm, and pyocyanin are the major determinants critical for chronic infections caused by *P. aeruginosa*[33]. In this work, SrpA was experimentally demonstrated to influence the activities of multiple cellular processes related to the main virulence factors of *P. aeruginosa*, namely T3SS, type IV pili, pyocyanin, swarming, cell shape, T6SS, chemotaxis, and biofilm. To further confirm the function of the *srpA* gene, a mutant strain completely deleted of the *srpA* gene (KO4) was generated (Supplementary Table 2). In cell motility assay, KO4 displayed similar phenotypes as the strain 1–9 (Supplementary Fig. 5), indicating that the *srpA* gene was fully inactivated in both strains.

In the *srpA* mutant, a number of known T3SS-related regulators are found significantly downregulated such as *tspR*, *suhB*, *fis*, and *vfr* or upregulated such as *rsmA* (Supplementary Data 1). Vfr is a global regulator that can fully restore T3SS gene expression by binding to the intergenic region between *exsB* and *exsA* and initiates *exsA* transcription from the second promoter[34–36]. TspR affects transcription of T3SS genes through a positive translational control of the ExsA and a negative transcription control of two small RNAs RsmY and RsmZ, which can sequester RsmA and inhibit T3SS expression[37]. Fis is an important regulator of *P. aeruginosa* virulence in murine acute pneumonia model, it can bind to the intergenic region between *exsB* and *exsA* in the *exsCEBA* operon, assisting the transcription elongation from *exsB* to *exsA*, maintaining ExsA at a certain level[38]. SuhB is an extragenic regulator playing a critical role in the translation of ExsA, it was also found to repress the transcription of *gacA*, and the downstream small regulatory RNAs, RsmY and RsmZ[39]. Additionally, a SrpA-binding site is found within the upstream region of *tspR*, hinting a direct control of the *tspR* expression by SrpA (Fig. 8). The above data suggested that SrpA plays a central role in the global regulation of T3SS and bacterial virulence (Fig. 10).

Multiple cellular systems contribute to swarming motility of *P. aeruginosa*, especially the presence of functional flagellar and

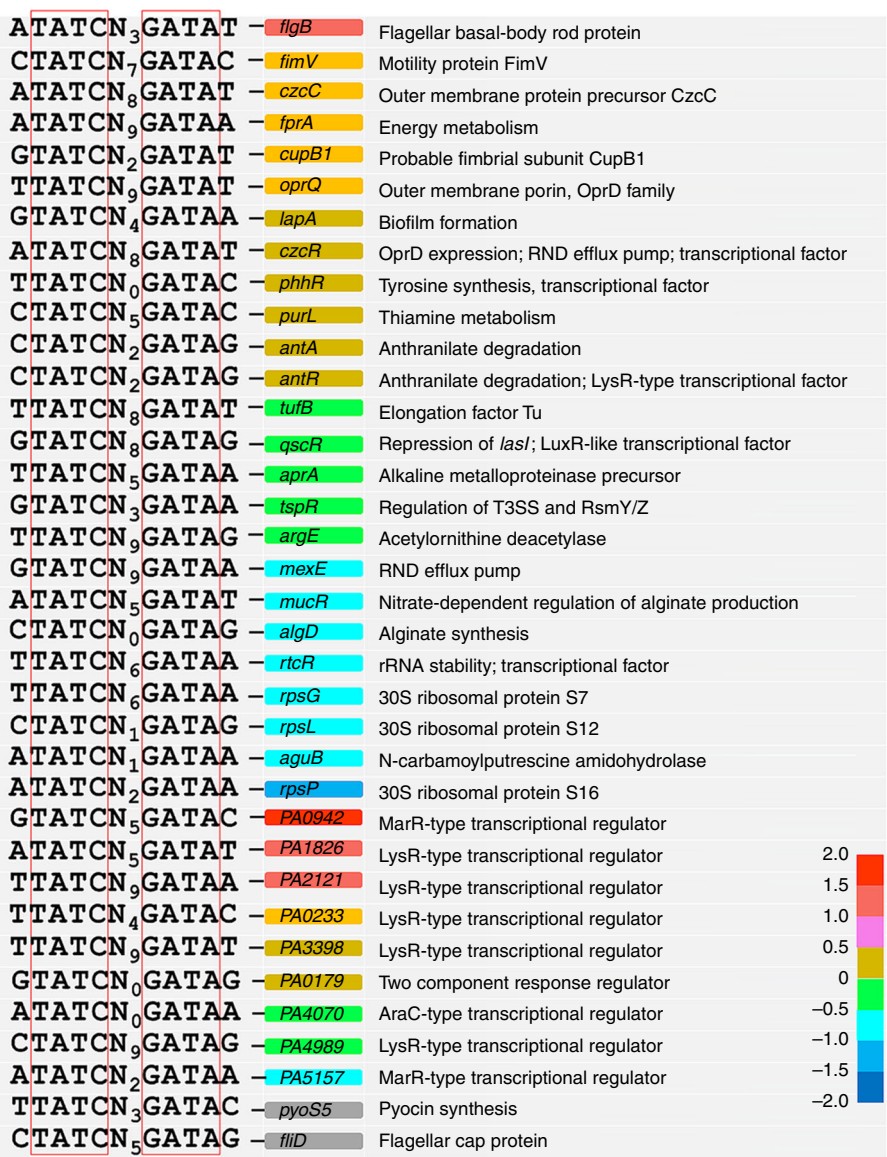

**Fig. 8** Identification of SrpA-binding sites in bacterial genome. The SrpA-binding sequences (5′-NTATCN(0,9)GATAN-3′) were searched across the draft genome sequence of PAK (http://www.pseudomonas.com/replicon/setmotif). Different colors depict the log2-transformed value of expression change folds of the indicated genes. Hypothetical regulators are represented with gene IDs, separated from the genes with known functions. No expression data is available for *pysS5* and *fliD*

type IV pili[40]. FlgM is a negative regulator of flagellin synthesis through sequestering the σ[28] factor FliA that is required for the flagellar biogenesis[41]. However, both the *flgM* and *flgF* genes, encoding the main component (flagellin) of flagella, are significantly downregulated in the *srpA* mutant. SrpA-binding sites are found in the promoters of the *flgB* and *fliD* genes, suggesting that SrpA can regulate these two genes directly. FlgB is a flagellar basal-body rod protein and its transcription is significantly increased in the *srpA* mutant (Supplementary Data 1 and Fig. 5). The transcription data of the flagellar genes obtained in this work further supports that swarming is a complex adaptive process responding to environmental signals[42].

FimV is required for type IV pili (Tfp)-mediated twitching motility in *P. aeruginosa*[43]. In strain 1–9, both *fimV* and *pilE* (encoding the major Tfp subunit protein) are significantly upregulated, while PilF, a lipoprotein essential for the Tfp biogenesis, and MreB, which is responsible for the transport of the Tfp proteins to polar locations[21], are both significantly downregulated. Hence the 1–9 mutants may not produce enough

functional Tfp, resulting in an attenuated twitching motility (Figs. 5 and 10). Additionally, the expression of *PA3804* is reduced in the *srpA* mutant (Supplementary Data 1). This gene encodes a putative cytoskeleton protein RodZ, which is essential for bacterial cell morphogenesis and required for assembly of the actin cytoskeleton MreB[44–46]. Taken together, the *mreB* operon and the *pilF-rodZ* operon may share a common regulatory mechanism via the SrpA to ensure a precise coordination between the multiple functional pathways. More experiments are needed to further reveal the underlying mechanisms.

RsmA is a carbon storage regulator blocking T6SS translation at the post-transcriptional level by binding to the 5′-UTR of messenger RNAs (mRNAs). Two downstream regulatory RNAs, RsmY and RsmZ, can sequester RsmA and antagonize its function[33, 47]. RetS represses H1-T6SS expression by downregulating the *rsmY* and *rsmZ* transcription through a two-component regulatory system GacA/GacS[33]. SuhB and TspR may affect T6SS activity through the transcriptional regulation of the small RNAs RsmY and RsmZ[37, 39]. The serine/threonine protein

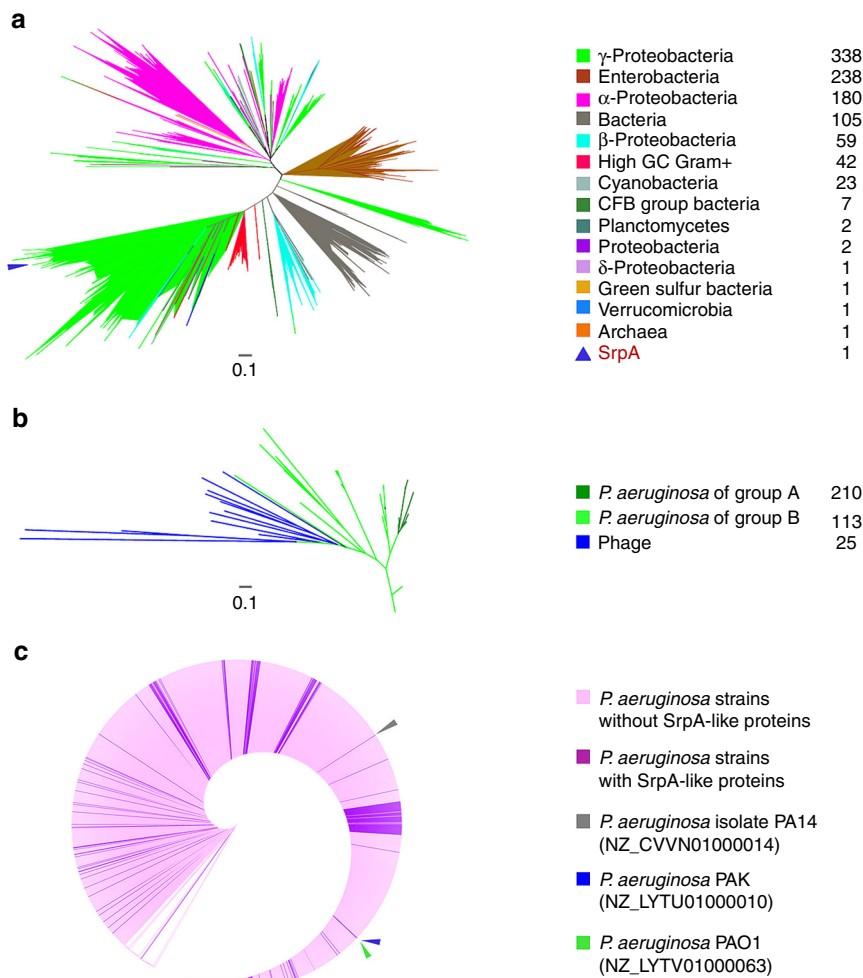

**a**

| | | |
|---|---|---|
| ■ | γ-Proteobacteria | 338 |
| ■ | Enterobacteria | 238 |
| ■ | α-Proteobacteria | 180 |
| ■ | Bacteria | 105 |
| ■ | β-Proteobacteria | 59 |
| ■ | High GC Gram+ | 42 |
| ■ | Cyanobacteria | 23 |
| ■ | CFB group bacteria | 7 |
| ■ | Planctomycetes | 2 |
| ■ | Proteobacteria | 2 |
| ■ | δ-Proteobacteria | 1 |
| ■ | Green sulfur bacteria | 1 |
| ■ | Verrucomicrobia | 1 |
| ■ | Archaea | 1 |
| ▲ | SrpA | 1 |

**b**

| | | |
|---|---|---|
| ■ | *P. aeruginosa* of group A | 210 |
| ■ | *P. aeruginosa* of group B | 113 |
| ■ | Phage | 25 |

**c**

| | |
|---|---|
| ■ | *P. aeruginosa* strains without SrpA-like proteins |
| ■ | *P. aeruginosa* strains with SrpA-like proteins |
| ■ | *P. aeruginosa* isolate PA14 (NZ_CVVN01000014) |
| ■ | *P. aeruginosa* PAK (NZ_LYTU01000010) |
| ■ | *P. aeruginosa* PAO1 (NZ_LYTV01000063) |

**Fig. 9** Prevalence of SrpA-like proteins. **a** Phylogenetic tree of the SrpA-like proteins ($n = 1000$) with the $E$-values lower than $5E−08$. Blue triangle stands for the SrpA protein. The scale bar represents the number of substitutions per site. **b** Phylogenetic tree of the SrpA-like proteins of *P. aeruginosa* and phages. The scale bar represents the number of substitution per site. Dark green branches stand for the SrpA homologs ($n = 210$) from the genome database of *Pseudomonas* with the $E$-value lower than $9.2E−17$ (group A). Green branches stand for the SrpA homologs ($n = 113$) in *P. aeruginosa* strains from the non-redundant protein sequences (nr) database with the $E$-value lower than 0.001 (group B). Blue branches stand for the SrpA homologs in phages from the non-redundant protein sequences (nr) database with the $E$-value lower than 0.001 ($n = 25$). **c** Phylogenetic analysis of the 2156 *P. aeruginosa* strains on the basis of the 16s rRNA gene sequence. Branch lengths are ignored in the phylogenetic tree circle. Pink lines stand for *P. aeruginosa* strains. Purple lines stand for the strains with SrpA homologs. Colored triangles stand for the reference strains PA14, PAK, and PAO1, respectively. The colored squares represent different groups as indicated

kinase PpkA and its cognate serine/threonine phosphatase PppA are encoded on a *tagQ1-tagR1-tagS1-tagT1-ppkA-pppA-tagF1-icmF1* gene operon of H1-T6SS. Both *ppkA* and *pppA* are upregulated as all other H1-T6SS genes in the *srpA* mutant (Supplementary Data 1). Activation of H1-T6SS requires the phosphorylation of Fha1 by the PpkA via four membrane-bound T6SS proteins TagQ, TagR, TagS, and TagT, while PppA downregulates T6SS activity through dephosphorylation of the phosphorylated Fha1[48, 49]. Combined together, these data indicate that SrpA plays an important role in the control of H1-T6SS gene expression via a number of regulatory pathways.

Biofilm formation is regulated by multiple regulatory systems. In the *srpA* mutant, the upregulated *rsmA* gene and the downregulated *rhlI* gene are consistent with the reduced biofilm production[14, 50]. Other factors may also influence biofilm synthesis in this strain. Four biofilm-associated genes *fis*, *lapA*, *algD*, and *mucR* were also found downregulated in strains 1–9, and the later three genes have SrpA-binding sites in their promoter regions (Fig. 8 and Supplementary Data 1). Fis is involved

in the regulation of biofilm formation via inhibition of the *lapF* expression by binding to its promoter[51]. The change of relative ratio between LapF and LapA may contribute to the reduction of biofilm formation[52]. The *algD* gene encodes a GDP-mannose dehydrogenase required for the synthesis of the alginate precursor GDP-mannuronic acid. The *mucR* gene encodes an enzyme with the GGDEF domain, a signature for diguanylate cyclases that synthesizes intracellular second messenger cyclic diguanylate (c-di-GMP). Both *algD* and *mucR* are subjected to the direct regulations by AlgR[53, 54]. Collectively, this work showed that SrpA plays roles in controlling biofilm formation, possibly through complicated regulatory networks (Fig. 10).

Our data show that the absence of SrpA leads to attenuated T3SS activity (Fig. 6b), reduced biofilm formation (Fig. 5), decreased twitching motility (Fig. 5), increased swarming motility (Fig. 5), enhanced T6SS activity (Fig. 5), reduced pyocyanin formation (Fig. 6a), and elevated chemotaxis (Fig. 5). Additionally, other virulence-related genes were also found downregulated in the strains 1–9 (Supplementary Data 1), such as the *rhlI* gene

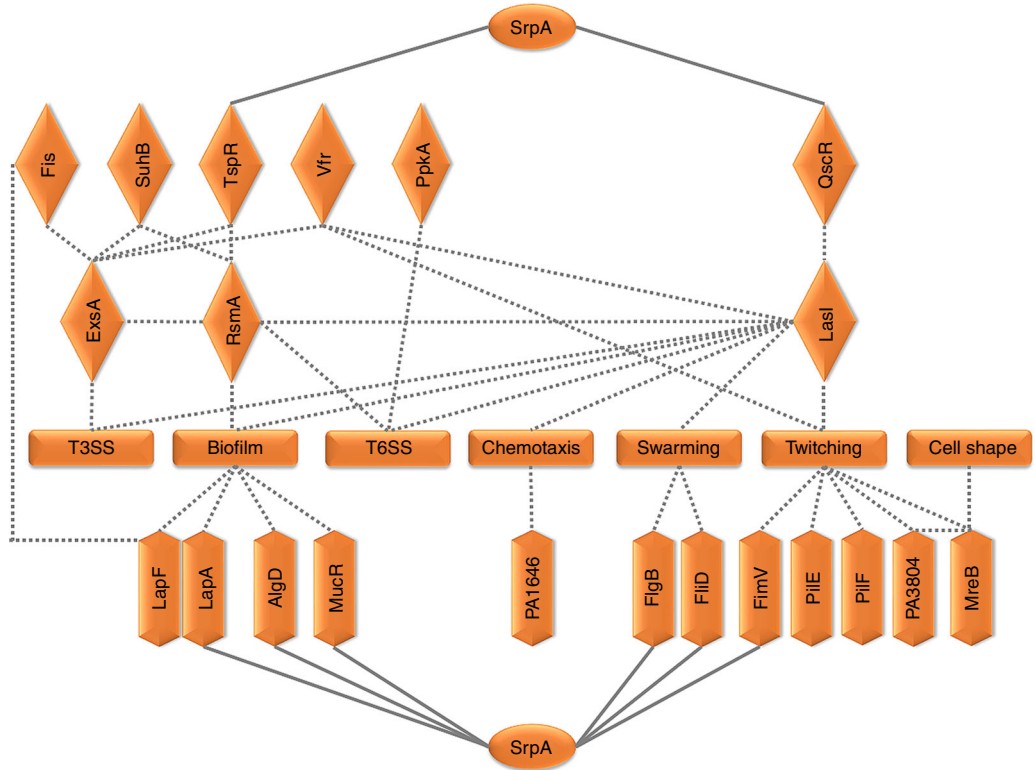

**Fig. 10** Schematic representation of the SrpA regulatory web in *P. aeruginosa*. The regulatory network includes the cellular processes and the genes significantly influenced by the SrpA regulator. It also includes the genes involved in the indicated cellular processes since they have SrpA-binding sites in their promoter regions. *PA1646* encodes a putative methyl-accepting chemotaxis transducer and highly expressed in the absence of SrpA. *PA3804* encodes a putative cytoskeleton protein RodZ required for the assembly of actin cytoskeleton MreB. Solid lines represent downstream genes with putative SrpA-binding site in their promoters. Dotted lines represent direct or indirect regulatory relationships that have been established previously. SrpA is depicted in oval textboxes. Regulators are depicted in diamond textboxes. Structural proteins or enzymes are depicted in elongated hexagon textboxes. Cellular processes are depicted in rectangle textboxes

controlling synthesis of pyocyanin, elastases, and rhamnolipids[55]. All these factors contribute to the virulence and pathogenicity of *P. aeruginosa*[56–58]. The outcome of *C. elegans* killing assay represents the overall attenuated feature of the *srpA* mutant.

*P. aeruginosa* is the dominant pathogen in chronic lung infections of CF patients. The CF isolates of *P. aeruginosa* are phenotypically different from the strains isolated from acute infections due to adaptions to the specialized niches of CF lungs with different challenges such as antibiotic chemotherapy and host inflammatory responses[59, 60]. Genetically, these CF isolates are hypermutable with significantly increased spontaneous mutation rates[61]. Phages have also been shown abundant in this special microecosystem of the CF lungs[62, 63]. The existence of the enormous phage population contributes greatly in cutting down the bacterial population densities through lysis[64]. Interestingly, phage selection force can be another important challenge to bacterial pathogens, and some phages have been confirmed to contribute to the conversion of mucoid phenotype in CF isolates of *P. aeruginosa*[65]. In our study, disruption of the *srpA* gene simultaneously confers phage resistance and numerous physiological alterations in the *P. aeruginosa*, suggesting that bacterial genes accounted for structural proteins, enzymes, or regulators could potentially be involved in phage infection process. Our data suggest that phage selection may be an important driving force for bacterial diversity in various ecological niches including that of CF lungs.

In summary, as a new member of small regulatory proteins, SrpA is required for the effective phage infection by controlling the transcription of the phage genes in *P. aeruginosa*. More importantly, SrpA is a novel key regulator controlling core cellular processes through direct or indirect regulation of dozens of bacterial genes. Further research is needed to understand the roles played by other SrpA-like proteins, which are widely present in bacteria.

## Methods

**Bacterial strains and growth conditions**. *P. aeruginosa* strain PAK-AR2 is a *purEK* deletion mutant of PAK, and both of them are sensitive to the phage K5 as characterized previously[10]. A mutant strain 1–9, resistant to the phage K5, had previously been isolated from a transposon Tn5G insertion mutant library of strain PAK-AR2[10]. *E. coli* OP50 was used as a control in worm killing assay. *E. coli* DH5α was used for plasmid construction and *E. coli* M15 for protein expression. Bacterial cultures were routinely incubated in LB medium supplemented with appropriate antibiotics at 37 °C. Bacterial strains, plasmid vectors, and recombinant plasmids used in this work are listed in Supplementary Table 2. Primers used in this work are listed in Supplementary Table 3.

**Identification of the Tn5G insertion mutant**. Adsorption rate of the phage K5 to *P. aeruginosa* strains was measured with an MOI (multiplicity of infection) of 0.001[66]. Inverse PCR was performed to identify transposon Tn5G insertion sites using a primer pair OTn1 and OTn2 (Supplementary Table 3). The obtained sequences were compared in the genome database (http://beta.pseudomonas.com/) to identify the insertion sites. The target gene with its own promoter was amplified and cloned into the vector pUCP18. The recombinant plasmid was introduced into strain 1–9 for complementation test. Phage sensitivity of the indicated *P. aeruginosa* strains were determined using $10^5$–$10^6$ phage particles for each spot. Protein similarity was predicted using the online software BLASTP program (https://blast.ncbi.nlm.nih.gov/Blast.cgi?PROGRAM=blastp&PAGE_TYPE=BlastSearch&LINK_LOC=blasthome) and the DNA-binding domain and the residues-binding DNA were displayed with the program Conserved Domains in the same window. SrpA homologs were selected with *E*-value lower than the threshold of $5E-08$ by searching non-redundant protein sequences (nr) database using PSI-BLAST (position-specific iterated BLAST) in

NCBI. Conserved amino acid residues were predicted from the top 1000 SrpA homologous sequences using an online tool WebLogo (http://weblogo.berkeley.edu/).

**Growth inhibition curves**. Overnight culture of strain 1–9 was co-cultivated with serial dilutions of the phage K5 at MOIs of 0, 1, and 10, respectively. Optical densities of cultures were measured at the wavelength of 600 nm at indicated time intervals during the incubation period.

**Southern blot analysis**. Bacteria-phage co-cultures were prepared by one-step growth experiment[66]. Strain 1–9, 1–9/pLLY1101, or PAK-AR2 was incubated with the phage K5 at an MOI of 1. Multiple 20-ml cultures were taken at 20 min intervals. Total genomic DNA was extracted from the collected cells according to the method described previously[67]. DNA samples were digested with the restriction endonuclease $EcoRV$ for southern blot analysis. Primers were designed to synthesize the probe corresponding to region from 2854 to 3373 bp of the K5 genome (Supplementary Table 3). The genomic DNA of PAK-AR2 and the phage K5 was used as negative and positive controls, respectively. DNA probe synthesis and DNA hybridization were conducted using the Detection Starter Kit II (Roche) according to the manufacturer's instruction.

**Real-time quantitative PCR**. Bacteria-phage co-cultures were prepared by one-step growth experiment[66]. PAK-AR2 or 1–9 was incubated in 40 ml LB medium until the cell density reached 0.8 at $OD_{600}$. The cells were collected and resuspended in 0.5 ml fresh LB medium and mixed with 0.5 ml of the phage K5 solution that resulted in an MOI of 0.01. After 1 min standing at room temperature, the mixture was centrifuged at 13,000×$g$ for 30 s. The collected cells were immediately resuspended in 100 ml LB and incubated at 37 °C with shaking (200 r.p.m.). Multiple 1-ml samples of the cultures were taken every 6 min. Total genomic DNA was extracted from the collected cells according to the described method[67] and used as template for measurement of phage genomic DNA by quantitative PCR. The phage RNA polymerase gene gp058 was selected as the target gene for quantification. The genomic DNA of PAK-AR2 and water were used as negative controls, while the genomic DNA of the phage K5 was used as the positive control. PCR was conducted using the GoTaq® qPCR Master Mix (Promega) with the indicated primers (Supplementary Table 3). All samples were tested in triplicates.

**Quantitative real-time reverse transcription PCR**. Bacteria-phage co-cultures were prepared by one-step growth experiment[66]. Strain 1–9 or PAK-AR2 was incubated with the phage K5 at an MOI of 0.01. Multiple 1-ml cultures were taken at every 6 min. The collected cells were subjected to total RNA extraction using a RNAiso Plus kit (Takara). Total RNA (1 μg per reaction) was subjected to reverse transcription using the GoScript™ Reverse Transcription System and the complementary DNA (cDNA) was subjected to real-time quantitative PCR analysis using the GoTaq® quantitative PCR Master Mix (Promega) with the indicated primers (Supplementary Table 3). Expression level of phage genes gp058, gp104, and gp105, encoding putative RNA polymerase, DNA polymerase I, and DNA polymerase II, respectively, was determined. All samples were tested in triplicates.

**Construction of transcriptional reporters**. The promoters of the RNA polymerase gene gp058, the DNA polymerase I gene gp104, the DNA polymerase II gene gp105, the major capsid protein gene gp056, the base plate protein gene gp071, and the ribonucleotide diphosphate reductase β-subunit gene gp128 were amplified with the indicated primers, respectively (Supplementary Table 3). The PCR products were cloned into the pDN19lacΩ plasmid, respectively. The resulting recombinant plasmids were introduced into PAK-AR2 and 1–9 by electroporation. The target gene expression levels were determined by measuring β-galactosidase activities of the transformants.

**Bacterial two-hybrid assay**. The srpA gene and the RNA polymerase gene gp058 were amplified with the indicated primers (Supplementary Table 3), respectively. The PCR products were cloned into the bait vector (pBT) and pray vector (pTRG), respectively. The interaction between the gene products was determined by measuring β-galactosidase activity.

**Electrophoretic mobility shift assay**. The srpA gene was amplified with the indicated primers (Supplementary Table 3). The PCR product was cloned into the plasmid pQE30 for overexpression of a His-tagged SrpA protein. The indicated promoters were amplified with the indicated primers (Supplementary Table 3), respectively. The PCR products were cloned into the plasmid pMD19 (Simple) (Supplementary Table 2), respectively. The cloned promoter fragments were then amplified using the fluorescence-labeled primers M13-47-cy3 and RV-M-cy3 (Supplementary Table 3). Purified His-tagged SrpA protein (0–7.42 μM) was incubated with the Cy3-labeled PCR products (0.33–1.03 μM) in 2× binding buffer (40 mM Tris-HCl (pH 7.5), 4 mM MgCl₂, 100 mM NaCl, 10% glycerol, 2 mM DTT, 0.2 mg/ml BSA, 0.02 mg/ml poly (dI-dC), and 1 mM EDTA) at 25 °C for 30 min. The mixtures were subjected to electrophoresis on a 5% native PAGE. The gel was visualized with the ImageQuant LAS 4000 (GE Healthcare Life Sciences, USA).

**Transcriptome analysis**. P. aeruginosa PAK-AR2 and 1–9 cells were grown to $OD_{600}$ of 0.8 before collecting. The collected cells were treated with RNAprotect Bacteria Reagent (Qiagen) and subjected to snap freezing in liquid nitrogen and delivered in dry ice to BGI (Beijing Genomic Institute) for transcriptome resequencing analysis (http://www.genomics.cn/en/index). Briefly, total RNA was extracted from bacterial cells and subsequently subjected to rRNA removing to obtain mRNA. Purified mRNA was fragmented and used as templates for cDNA synthesis. The cDNA libraries were sequenced using Illumina HiSeq™ 2000. The genome of P. aeruginosa PAO1 (NC_002516.2) was used as reference for annotation. A total of 4,355,305 reads matched to the referenced genome in the sample of PAK-AR2, and 3,544,484 reads in the sample of 1–9. The differentially expressed genes were determined between PAK-AR2 and 1–9 with the standards of false discovery rate (FDR) ≤0.001, fold change |log2Ratio| ≥1. KEGG database was used to classify the genes with significant differential expression based on their functions. The online software DAVID Bioinformatics Resources 6.8 was used to convert gene ID to ENTREZ_GENE_ID (https://david.ncifcrf.gov/content.jsp?file=functional_classification.html). The Pathway Enrichment Analysis tool Omicshare was used to classify genes at the level of KEGG_B_class (http://www.omicshare.com/tools/Home/Soft/pathwaygsea?l=en-us). Genes with significantly differential expression were grouped into the core cellular processes based on their annotations described in the microbiome analysis.

**Motility assays**. For swimming assay, bacterial culture was grown to an $OD_{600}$ of 0.6. One microliter of the culture was spotted onto the semi-solid medium consisting of 2% tryptone, 1% K₂SO₄, 0.14% MgCl₂, 0.002% Tween 20, and 0.2% agarose. After incubation at 37 °C for 24 h, diameter of the swimming zones was measured. For swarming assay, semi-solid medium with 0.3% agarose was used, and the dendritic structure was observed. For twitching motility assay, fresh colonies were stab-inoculated through the 1% L-agar layer. After incubation at 37 °C for 24 h, the plates were fixed and stained with 0.25% Coomassie brilliant blue R-250 solution to visualize the twitching motility zones.

**Chemotaxis assay**. For chemotaxis assay, a 200 μl yellow pipette tip was filled with 50 μl exponential culture ($OD_{600}$ of 0.6); 1 ml 0.5% casein hydrolysates was then drawn into a 10 ml sterile syringe with air bubbles expelled; and then the needle was dipped into the culture in the prefilled yellow tip. After incubation at room temperature for 30 min, the mixture was transferred from the syringe into a 1.5 ml microcentrifuge tube and subjected to plating for bacterial enumeration.

**Biofilm formation assay**. For biofilm assay, 20 μl overnight culture was transferred into a well containing 200 μl fresh medium (0.33% peptone, 0.33% NaCl, and 0.17% yeast extract) in a flat bottom 96-well microplate. After incubation at 37 °C for 48 h without shaking, cultures were gently removed and the wells were washed with water for three times. The plates were stained with 0.1% crystal violet solution for 15 min. Bound crystal violet was extracted from the stained biofilm with 95% ethanol for 15 min and measured the absorbance at the wavelength of 595 nm.

**Pyocyanin production assay**. Bacterial cultures were incubated in LB medium for 24 h and supernatants were collected by centrifugation. One mililiter supernatant was extracted with 0.5 ml of chloroform, and then the pyocyanin-chloroform solution (0.4 ml) was extracted again with 0.3 ml 0.2 M HCl. Optical density of the resulting pink solutions was measured at the wavelength of 520 nm. Quantities of the pyocyanin (μg/ml) was determined by multiplying the optical density by 32.01 according to the method described previously[68].

**Western blot analysis**. Overnight cultures of the indicated bacterial strains were transferred into fresh LB containing 5 mM EGTA at 1% inoculum and continuously incubated at 37 °C for 3 h. The supernatants were collected and concentrated with the addition of 15% trichloroacetic acid (TCA). The samples with equivalent bacterial cells were subjected to electrophoresis on 12% SDS-PAGE. PVDF-PLUS membrane was used for the protein transfer. Rabbit antiserum specific to ExoS was used to detect T3SS effectors at 1:1000 dilution[39]. Hybridization was performed by using 1 mg/ml peroxidase-conjugated anti-rabbit IgG (H + L) (Promega) at 1:2000 dilution, the signal was developed with ECL Plus kit (GE Health).

**Worm killing assay**. Worm killing assay was performed as described previously[69]. The wild-type strain C. elegans N2 was cultured on nematode growth medium (NGM) agar plates coated with E. coli OP50 cells at 20 °C (Supplementary Table 2). For synchronization, cultured C. elegans was collected by rinsing the plates with M9 buffer and collected by centrifugation at 2500×$g$ for 3 min. After repeatedly washing with M9 buffer, C. elegans was lysed by the addition of 20% alkaline hypochlorite solution and shaking for 4 min. C. elegans eggs were collected by centrifugation and resuspended in 30 ml M9 buffer. Eggs was incubated with shaking at a speed of 70 r.p.m. at 20 °C for 10 h. Larvae were transferred onto the NG plates coated with E. coli OP50 cells and cultured for 48 h. The synchronized worms were transferred to the PGS plates (>30 per plate), which was coated with 50 μl overnight cultures of the indicated P. aeruginosa strains and incubated at

37 °C. The number of worms were counted at every 12 h. The assay was conducted in triplicate for each strain.

**Antibacterial activity assay.** To test T6SS-mediated antibacterial activity, PAK-AR2, 1–9, and 1–9/pLLY1101 were used as attacker strains while *E. coli* DH5α carrying a *gfp* gene was used as the target strain (Supplementary Table 2). Bacteria were grown to exponential phase (OD$_{600}$ = 0.6) and the attacker cells were mixed with target cells at an initial ratio of 1:1. Cells were collected by centrifugation and transferred on L-agar plates for incubation. At 30 min time intervals, cells were rinsed off with 0.9% NaCl solution. The DH5α-GFP cells were counted with the Accuri C6 flow cytometer (BD). The assay was conducted in triplicate for each strain at every time points. Additionally, the mixed cells were visualized under a fluorescent microscopy Olympus BX53 (Olympus).

**Scanning electron microscopy.** Scanning electron microscope SU-1510 (Hitachi High-Technologies Global) was used to observe cell shapes. Bacterial cells at exponential phase (OD$_{600}$ = 0.6) were collected by centrifugation and washed once with 0.9% NaCl. The collected cells were treated with 2.5% glutaraldehyde for overnight at 4 °C. The fixed cells were collected and washed once with 0.9% NaCl. Then the cells were treated sequentially with increasing ethanol solutions, including 30, 50, 70, 80, 90, and 100%, for 15 min of each dehydrating step. The samples were dropped onto 0.25 × 0.25 cm coverslips for drying at room temperature for 4 h and coated with gold. Cell images were recorded at 10,000 × magnification. The length and width of the cells were analyzed with the software ImageJ (https://imagej.nih.gov/ij/download.html). Forty cells of each indicated strain were measured for the analysis.

**Construction of the *srpA* gene deletion mutant.** The upstream and downstream fragments of the *srpA* gene were amplified. After ligation, the ligated molecule was amplified again and the PCR product was cloned into pEx18Gm. The resulted plasmid was electroporated in *E. coli* S17-1, and then introduced into the strain PAK by conjugation. The *srpA* gene deletion mutants were selected by using L-agar medium supplemented with 8% sucrose. The bacterial strains and plasmids used were listed in Supplementary Table 2 and the primers used were listed in Supplementary Table 3.

**Identification of SrpA-binding sites.** SrpA-binding sites were identified by searching DNA motif in the genome database of *Pseudomonas* (http://www.pseudomonas.com/replicon/setmotif). The palindromic nucleotide sequence 5′-NTATCN(0,9)GATAN-3′ was used as query for sequence similarity searching against the draft genome of PAK. Also, the palindromic sequence was searched through the genome sequence of the phage K5 using an online tool (http://meme-suite.org/tools/mast).

**Phylogenetic tree construction of SrpA-like proteins.** The distribution of SrpA homologs in microorganisms were studied by searching non-redundant protein sequences (nr) database using PSI-BLAST (Position-Specific Iterated BLAST) in NCBI. Total 1000 homologous sequences were selected with *E*-value lower than the threshold of 5*E*−08. The distance tree was generated using BLAST pairwise alignments in NCBI. The phylogenetic tree of SrpA-like proteins from *P. aeruginosa* and phages with *E*-value lower than 0.001 was constructed using a software MEGA7. SrpA-like proteins from the genome database of *Pseudomonas* were also included in the analysis. The distribution of SrpA homologs in *P. aeruginosa* was analyzed by construction of a phylogenetic tree of the 16s rRNA gene sequences from 2156 strains by searching the genome database of *Pseudomonas* using a software MEGA7. The phylogenetic trees were modified using an online software (http://itol.embl.de/).

**Statistical analysis.** One-way ANOVA was used for comparing statistical difference between the groups of experimental data.

**Data availability.** The RNA sequence data have been deposited in the GEO database of NCBI with the accession code GSE112354. Other relevant data supporting the findings of the study are available in the article and its Supplementary Information files, or from the corresponding authors upon request.

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

## Acknowledgements
This work is supported by The National Natural Science Foundation of China (grant nos. 31370205 and 30970114).

## Author contributions
J.Y. performed the bioinformatic analysis and experiments and wrote the manuscript. X.P. performed bioinformatic analysis. L.S. and X.Y. contributed to southern blot and EMSA analyses. L.L., X.C., Z.J., X.Z., and M.G. carried out plasmid constructions and measured β-galactosidase activity of reporter genes. Z.H. and Z.F. constructed and characterized the srpA deletion mutant. Z.R. contributed to the analysis design. S.J. and W.W. contributed to the analysis design and data interpretation. H.Y. designed the experiments and wrote the manuscript. All authors reviewed the manuscript.

## Additional information

**Competing interests:** The authors declare no competing interests.

