## [Peer Review File · Nature Communications]

Reviewers' comments:

Reviewer #1 (Remarks to the Author):

This work of Jiajia You and others presents a novel regulatory protein SrpA of *Pseudomonas aeruginosa* strain PAK that appears to have a broad regulatory influence on the phenotype of strain PAK and is essential to infection with phage K5. The manuscript presents a thorough investigation of the phenotype of a SrpA deletion mutant that reveals pleiotropic effects affecting various virulence-related traits, e.g. biofilm formation and expression of Type III and Type VI secretion systems. In addition, the SrpA protein was proven to directly influence the transcription of various genes, which interestingly includes genes of the phage genome, and a consensus binding motif was identified. This clearly characterizes SrpA as a DNA-binding regulatory protein.

Unfortunately, the description of the methods and the interpretation of the results have weaknesses that render the results difficult to reproduce and interpret. This is a pity, because the data do point towards a potentially important role of SrpA but at least I found it difficult to assess its actual importance. I'll try to point out the most important aspects of the manuscript that I think could be improved:

1) Identification of the *srpA* gene: the results section describes the gene to be situated "between the genes PAK_03825 and PAK_03826", which is quite vague. While the *Pseudomonas* database (pseudomonas.com) does not display any coding sequence between the two genes, one can find an "XRE family transcriptional regulator" in the NCBI nucleotide collection in the vicinity of gene PAK_03825. Referring to this gene directly would make it much easier for the reader to follow the results and actually identify the gene in question themselves.

I was not able to deduce how the authors came up with the protein name SrpA. The way it is written suggests that this name existed in the database but I was not able to find any homolog with this name. A BLASTP search in the non-redundant database only yields "XRE family transcriptional regulator" hits without short names. Did the authors name the gene themselves? What does it stand for?

2) Role of SrpA: the phenotype of the 1-9 mutant clearly shows broad pleiotropic effects of the SrpA deletion. However, I think that describing SrpA as a "global regulator controlling the core cellular processes (L. 637)" is misleading. The described effects on various virulence factors are observed very often when regulatory genes are deleted and represent mostly a switch between a motile lifestyle usually connected to acute infections and virulence and a biofilm-lifestyle that is less virulent but still connected to chronic infections. Most of this regulation is most likely indirect and mediated via well-known global regulators like Vfr or the quorum sensing systems that are shown to directly influence hundreds of promoters. The direct SrpA regulon is apparently comparatively small compared to those global regulators. I'd argue that much of the pleiotropic effect of SrpA is in fact indirect.

3) "Co-evolution" of SrpA and phage K5: the authors argue that "data indicate that phage selection may be an important driving force for bacterial diversity (L. 631)" and that SrpA related genes "help bacteria to adapt to the challenge by phage through genetic mutations [of] the *srpA* gene (L. 643)". It is a common hypothesis that phages are a major driving force of bacterial evolution but still I do not really see how the presented data contribute to this view. The K5-resistant strain PAK-AR2 clearly has a pronounced phenotype that will probably largely diminish its fitness in competition with other strains unless a strong selective pressure by such phages is present. Doesn't this make the SrpA deletion a very unlikely resistance determining mutation *in vivo*? Especially if other phages are present that likely use other pathways to infect the bacteria? The presented data do not really allow to test such hypothesis in the context of natural selection and such statements should be made with caution.

4) SrpA deletion: Where did the transposon insert into the *srpA* gene in strain 9-1? Transposon mutagenesis is a “quick-and-dirty”-method to knock out genes that may lead to unexpected results. Since the original gene is still present a partially active version of SrpA might still be present that has a different phenotypic effect than a clean deletion (i.e. removing the gene completely).

5) SrpA protein structure: it is not clear, how the consensus in Fig. 1b was generated. Which sequences were included? How do we know, which residues bind the DNA (red arrows)? Please indicate the method or literature reference.

6) SrpA homologs: Fig. 9 suggests that SrpA is a widely distributed gene with homologs all over the microbial world. However, I was not able to find any homologs in the two model strains PAO1 and PA14 by BLASTN or BLASTP. How do the authors explain this? I think it is strange since the binding motif seems to be there and conserved (I've checked the *mexE* promoter, which carries the motif in PAO1 for example). Fig. 9 would be greatly improved if strains that do not carry SrpA homologs would be included to really give an estimate about the abundance of this gene, especially within the species *P. aeruginosa*.

7) Strain PAK-AR2: what is the rationale behind using a *purE*K deletion mutant in this context? The PAK wild type is clearly shown to display an even more pronounced phenotype (pyocyanin production is obviously affected but PAK-AR2 seems to be pyocyanin deficient, Fig. 6)

8) Transcriptome analysis: why was the genome of PAO1 used as a reference instead of the also available and annotated genome of the parent strain PAK? *P. aeruginosa* is known to have a mosaic genome structure with large variable regions that differ between the strains. Many genes of PAK are most likely missed in the analysis, which surpasses the advantage of using a completed vs. a draft genome.

9) Statistics: the statistical methods are not properly described. Fig 1 and 2 lack any statistical analysis (comparing the curves) and the p-values in Figs. 3, 5, 6, 7 are presented without revealing the statistical test used.

10) Language: the language is of varying quality. Some sections contain grammatical errors that make the text more difficult to read. I suggest having the text checked by someone who's proficient in English (not a native speaker myself, but I still found it distracting sometimes...).

Figures:

11) Fig 1: how was the consensus determined? How do we know about the red arrows?

12) Fig 2: Please indicate what is in lanes 1-4.

13) Fig 4: This one does not show significant genes, only the selected pathways, or does it? It'd be probably better to present a bar chart of different pathways with the amount of up- and downregulated genes. The current figure is hard to interpret and colors are difficult to distinguish.

14) Fig 9: I already commented on 9a. With 9b, I do not understand, why $E < 0.001$ was chosen for the BLAST search. This should result in many false positive homologs, since at this many low-level sequence similarities should show up that most likely do not represent homology.

15) Fig 10: This is very counterintuitive, since the colors indicate the effect of SrpA-deletion but the graph suggest the regulative effect of SrpA, which is the exact opposite. Some genes are shown as regulated that have no significantly differential expression in Table S3, e.g. *qscR* and *lasI*.

Reviewer #2 (Remarks to the Author):

In this manuscript, the authors identify a bacterially-encoded protein (SrpA) that functions as a regulator of phage infection and a number of host processes including motility, interspecies interactions, and cell shape. The breadth but not depth of this gene's distribution throughout the Pseudomonads and amongst other bacteria suggest horizontal transfer and perhaps particular function selected to increase resistance to phage while simultaneously increasing competitiveness against other microbial community members. This is an interesting story with strong supporting data. I have only a few comments/questions.

1) The authors base much of their reasoning, in terms of regulatory pathway alterations for their phenotypes, on work done in *P. aeruginosa* strains PAO1 and PA14. However, PAO1 and PA14 do not possess an ortholog of SrpA (personal search and the authors' Table S5). The authors should comment on this and how well they expect the SrpA-dependent regulation in strain PAK will translate to other *P. aeruginosa* strains. Optimally, the *srpA* gene should be expressed in PAO1 or PA14 and a few simple phenotypes examined (swarming and twitching) or, at very least, determine the conservation of the SrpA binding site in front of similar genes in one of these other strains.

2) Cell shape alterations are often pleiotropic and could explain the twitching motility, for instance, as longer cells tend to associate better and lead to more efficient twitching. Perhaps I missed it, but this should be a caveat to be included.

3) Fig3d: For the EMSAs, where is the rest of the band shifting as SrpA is added for Pgp058? Presumably to higher order structures. It would be useful to show the entire EMSA image for just this promoter region.

4) Writing quality: The Results section is reasonable and the Discussion section borderline, but the Introduction has a number of grammatical issues and needs heavy copy editing.

Reviewer #3 (Remarks to the Author):

You et al. NCOMMS-17-24307

Summary This manuscript describes the identification and subsequent analyses of SrpA, which controls both phage infectivity and core cellular processes in *Pseudomonas aeruginosa*. Overall, this is an interesting study but the authors should consider the following issues:

Review criteria

a. What are the major claims of the paper?

- The authors identified a bacterial encoded protein, SrpA, which can regulate both phage and bacterial genes.
- It is claimed that SrpA can monitor core cellular processes in response to phage infection and environmental signals, providing insights into other such proteins that are widespread in bacteria without any known function.

b. Are the claims novel and will they be of interest to others in the community and the wider field?

- Yes I feel the information conveyed in Fig. 9 could be of broad interest to the community and wider field. It is quite interesting that these proteins are widespread.

- I feel the paper would have been more impactful if the authors could better tie in the biology of the phage with the list of bacterial genes that are co-regulated by SrpA. Why does SrpA regulate this subset of bacterial genes and how is this important for the phage life cycle?

c. Is the work convincing?

- The manuscript is a bit challenging to read and could use English language editing. Significant editorial work throughout seems necessary.
- Fig. 2b, label the top of the gel as time of DNA harvest (minutes). What is in lane 1 and 3?
- Data in Fig. 3a is redundant with what is shown in Fig. 2 d-f. Why are both necessary? Also, the fusions being used to selected genes in Fig 3b are not labeled. Are they same as those being evaluated in Fig. 3c?
- I don't feel that the data described in Fig. 3 and Fig. S1 justify the statement that SrpA and the phage RNA polymerase possibly co-regulate transcription of phage genes (line 350 and title on line 338). If SrpA only controls expression of the phage polymerase then the effects on these other genes would be indirect. Indirect effects also seem to be supported by the EMSA experiment in Fig. 3d that shows SrpA only binds to gp058 promoter DNA. The key experiment would be to uncouple phage RNA polymerase expression from SrpA and then evaluate whether these phage genes still require SrpA for expression.
- Fig. 3e, are the numbers shown relative to the start of gp058 transcription?
- Fig. 4. I might suggest using an additional color to represent those phage K5 genes that were also found to be regulated in these two strains. It would appear these are all host (bacterial genes).
- Fig. 6. I found it strange that the impact of purEK mutation present in strain PAK-AR2 was only investigated for pyocyanin production and virulence. What is the rationale for this and what is the explanation for this phenotype? This seems dropped in the manuscript and the AR2/1-9 vs. PAK/C11-1 comparisons are only selectively used throughout the paper. Are the cell morphology differences described in Fig. 7d due to SprA or purEK? The length differences are not fully restored upon complementation.
- Fig. S2, these data are not convincing without appropriate controls including competing DNA molecules to show specificity. Why do the DNA binding activities of SrpA change so much for gp053 and gp125 at 40 ng/ul? What is the SrpA molar concentration used here?

d. Appropriateness and validity of the statistics.

- Most seem appropriate but I don't understand how there is a statistically significant difference in length between the strains being evaluated in panel 7d. The standard errors seem to overlap yet the p value is <0.001 .

e. Signatures if desired. Daniel Wozniak

Responses to Referees' Comments

Reviewer #1 (Remarks to the Author):

This work of Jiajia You and others presents a novel regulatory protein SrpA of *Pseudomonas aeruginosa* strain PAK that appears to have a broad regulatory influence on the phenotype of strain PAK and is essential to infection with phage K5. The manuscript presents a thorough investigation of the phenotype of a SrpA deletion mutant that reveals pleiotropic effects affecting various virulence-related traits, e.g. biofilm formation and expression of Type III and Type VI secretion systems. In addition, the SrpA protein was proven to directly influence the transcription of various genes, which interestingly includes genes of the phage genome, and a consensus binding motif was identified. This clearly characterizes SrpA as a DNA-binding regulatory protein.

Unfortunately, the description of the methods and the interpretation of the results have weaknesses that render the results difficult to reproduce and interpret. This is a pity, because the data do point towards a potentially important role of SrpA but at least I found it difficult to assess its actual importance. I'll try to point out the most important aspects of the manuscript that I think could be improved:

- 1) Identification of the *srpA* gene: the results section describes the gene to be situated "between the genes PAK_03825 and PAK_03826", which is quite vague.

While the *Pseudomonas* database (pseudomonas.com) does not display any coding sequence between the two genes, one can find an "XRE family transcriptional regulator" in the NCBI nucleotide collection in the vicinity of gene PAK_03825.

Referring to this gene directly would make it much easier for the reader to follow the

results and actually identify the gene in question themselves.

In a recently updated draft genome of *Pseudomonas aeruginosa* PAK (<http://beta.pseudomonas.com/>), a gene corresponding to our *srpA* gene has been annotated as *Y880_RS18270* on the assembly GCF_000568855.1: scaffold00004. This information has been updated in the revised manuscript.

I was not able to deduce how the authors came up with the protein name SrpA. The way it is written suggests that this name existed in the database but I was not able to find any homolog with this name. A BLASTP search in the non-redundant database only yields “XRE family transcriptional regulator” hits without short names. Did the authors name the gene themselves? What does it stand for?

SrpA is a putative DNA binding protein without annotation and the name is the abbreviation of the first letters (underlined) of the words “small regulatory protein”. This information has been added in the revised manuscript.

2) Role of SrpA: the phenotype of the 1-9 mutant clearly shows broad pleiotropic effects of the SrpA deletion. However, I think that describing SrpA as a “global regulator controlling the core cellular processes (L. 637)” is misleading. The described effects on various virulence factors are observed very often when regulatory genes are deleted and represent mostly a switch between a motile lifestyle usually connected to acute infections and virulence and a biofilm-lifestyle that is less virulent but still connected to chronic infections. Most of this regulation is most likely indirect and mediated via well-known global regulators like Vfr or the quorum sensing

systems that are shown to directly influence hundreds of promoters. The direct SrpA regulon is apparently comparatively small compared to those global regulators. I'd argue that much of the pleiotropic effect of SrpA is in fact indirect.

We agree with the reviewer's opinions. Although SrpA has been found to bind directly to 66 promoter regions and 15 of them control regulatory genes, more investigations are needed to confirm these observations. Accordingly, it may be more appropriate to replace the word 'global' with 'key'. Such changes have been made throughout the manuscript.

3) "Co-evolution" of SrpA and phage K5: the authors argue that "data indicate that phage selection may be an important driving force for bacterial diversity (L. 631)" and that SrpA related genes "help bacteria to adapt to the challenge by phage through genetic mutations [of] the *srpA* gene (L. 643)". It is a common hypothesis that phages are a major driving force of bacterial evolution but still I do not really see how the presented data contribute to this view. The K5-resistant strain PAK-AR2 clearly has a pronounced phenotype that will probably largely diminish its fitness in competition with other strains unless a strong selective pressure by such phages is present. Doesn't this make the SrpA deletion a very unlikely resistance determining mutation in vivo? Especially if other phages are present that likely use other pathways to infect the bacteria? The presented data do not really allow to test such hypothesis in the context of natural selection and such statements should be made with caution.

We agree with the reviewer's comments and delete the sentence "These genes will help bacteria to adapt to the challenge by phage through mutations of the *srpA* gene".

3) SrpA deletion: Where did the transposon insert into the *srpA* gene in strain 9-1? Transposon mutagenesis is a “quick-and-dirty”-method to knock out genes that may lead to unexpected results. Since the original gene is still present a partially active version of SrpA might still be present that has a different phenotypic effect than a clean deletion (i.e. removing the gene completely).

The *srpA* gene is 255 bp in length including the start and stop codons and the Tn5G transposon was inserted between the 37th and 38th nucleotide of the *srpA* gene. The details have been added in the first paragraph of the results section. As shown in Fig. 1b, the DNA binding domain is located between the 17 aa and 71 aa of the SrpA protein. It's unlikely that the remaining 12 N-terminal amino acids of the SrpA protein retains the DNA binding function as the full length SrpA protein.

As suggested by this reviewer, we have generated a mutant with the complete deletion of the *srpA* gene, called KO4. The mutant has been characterized and the relevant data are presented in the supplemented figure S5. The newly generated *srpA*⁻ deletion mutant displays similar phenotypes as the mutant with the *srpA* gene disrupted by Tn5G transposon.

5) SrpA protein structure: it is not clear, how the consensus in Fig. 1b was generated. Which sequences were included? How do we know, which residues bind the DNA (red arrows)? Please indicate the method or literature reference.

Protein similarity was predicted using the online software BLASTP by searching the

GenBank of NCBI. The DNA binding domain of SrpA and the amino acids binding the DNA sequence were displayed by the program Conserved Domains in the same window. Conserved amino acids were predicted using an online tool WebLogo (<http://weblogo.berkeley.edu/>). One-thousand SrpA homologs with E value lower than $5E-08$ were included in the assay. A more detailed description has been added in the materials and methods section and the results section of the revised version.

6) SrpA homologs: Fig. 9 suggests that SrpA is a widely distributed gene with homologs all over the microbial world. However, I was not able to find any homologs in the two model strains PAO1 and PA14 by BLASTN or BLASTP. How do the authors explain this? I think it is strange since the binding motif seems to be there and conserved (I've checked the *mexE* promoter, which carries the motif in PAO1 for example). Fig. 9 would be greatly improved if strains that do not carry SrpA homologs would be included to really give an estimate about the abundance of this gene, especially within the species *P. aeruginosa*.

This is the most intriguing part of the SrpA story. Although widely present in other bacterial species, only a small portion of *P. aeruginosa* strains carry the SrpA homologs, not including the reference strains PAO1 and PA14. The genomic context of PAK shows that the *srpA* gene is located between two genes, one encoding the DNA polymerase subunits gamma/tau (*Y880_RS18245*) and the other encoding the DNA ligase (*Y880_RS18290*). The homologs of these two genes are found in both PAO1 and PA14 (Fig. S4). The genome of PAK has an extra fragment of about 7.2 kb inserted between these two loci. This inserted fragment includes the *srpA* gene, several hypothetical protein encoding genes, and a putative integrase gene. The

genomic context indicates that this fragment was probably obtained via horizontal gene transfer which may even depend on the integrase. The information has been added in the results section of the revised manuscript (Fig. S3).

As suggested by this reviewer, the abundance of the *srpA* gene in the species of *P. aeruginosa* was estimated by searching the genome database of *Pseudomonas*. Totally 210 SrpA homologs with a threshold e-value lower than $9E-17$ were identified from 2156 strains, about 9.74%. The top 7 SrpA homologs are the same in the two data sets, one set of 210 SrpA homologs from the genome database of *Pseudomonas* and the other of 120 SrpA homologs from the non-redundant protein sequences (nr) database using PSI-BLAST (Position-Specific Iterated BLAST) in NCBI. These changes have been added to modify the Figure 9b. A phylogenetic tree based on the 16S rRNA gene sequences was generated and the strains with SrpA homologs were labeled (Figure 9c). The Figure 9c showed that the SrpA homologs were widely distributed with some clustered in several major clades in *P. aeruginosa* strains.

The core of the SrpA binding site is an 8 bp perfect palindromic structure. It is relatively simple and can be found in high frequencies in bacterial genomes. To test whether SrpA can function in PAO1 or PA14, recombinant strains were generated which overexpressed the *srpA* gene, then subjected cell motility tests. SrpA can reduce swarming capability, enhance twitching motility, but no effect on swimming, consistent with those observed in the PAK background as well as its derivatives (Fig. S4).

7) Strain PAK-AR2: what is the rationale behind using a *purE*K deletion mutant in

this context? The PAK wild type is clearly shown to display an even more pronounced phenotype (pyocyanin production is obviously affected but PAK-AR2 seems to be pyocyanin deficient, Fig. 6)

The strain 1-9 was selected from a collection of the mutants resistant to phage K5. Plasmid conjugation was used to introduce Tn5G into PAK for insertion library construction. However, it was difficult to completely remove *E. coli* donor cells from the mixture, thus the PAK derivative strain PAK-AR2, which harbors a spectinomycin/streptomycin resistance gene (Ω fragment), was used as a recipient. The antibiotics selection enabled us to get rid of the *E. coli* donor cells while selecting pure *P. aeruginosa* mutants.

When pyocyanin synthesis was first analyzed, no significant difference was observed between PAK-AR2 and 1-9. One possibility was that the *purEK* deletion may affect pyocyanin synthesis and it was confirmed by including two more strains PAK and C11-1 (the *srpA*⁻ mutant of PAK) (Fig. 6a). This part has been added in the revised version.

8) Transcriptome analysis: why was the genome of PAO1 used as a reference instead of the also available and annotated genome of the parent strain PAK? *P. aeruginosa* is known to have a mosaic genome structure with large variable regions that differ between the strains. Many genes of PAK are most likely missed in the analysis, which surpasses the advantage of using a completed vs. a draft genome.

We agree with the reviewer's opinions that some genes are possibly missed in the

analysis. However, it is hard to predict the exact gene numbers missed in the analysis since the gene numbers differ greatly between the assembled genomes of the two PAK strains (<http://beta.pseudomonas.com/>). *Pseudomonas aeruginosa* PAK (Assembly GCF_000568855.1) has 448 contigs with 6067 annotated genes and 614 partial genes, while *Pseudomonas aeruginosa* PAK (Assembly GCF_000408865.1) has 26 contigs with 5888 annotated genes and 26 partial genes. We have little information about the genomic homology of the PAK strain used in our work to that of the aforementioned PAK strains. Additionally, the draft genomes of PAK strains are not as well annotated as PAO1. In our transcriptome analysis, 5430 genes were identified using the PAO1 genome as the reference sequence, indicating the majority of the PAK genes were included in the RNAseq analysis.

9) Statistics: the statistical methods are not properly described. Fig 1 and 2 lack any statistical analysis (comparing the curves) and the p-values in Figs. 3, 5, 6, 7 are presented without revealing the statistical test used.

ANOVA (analysis of variance) was used to perform statistics analyses in our work and the information has been added in the materials and methods section as well as the figure legends.

10) Language: the language is of varying quality. Some sections contain grammatical errors that make the text more difficult to read. I suggest having the text checked by someone who's proficient in English (not a native speaker myself, but I still found it distracting sometimes...).

As suggested, the manuscript has been carefully edited by a colleague who is a native English speaker.

Figures:

11) Fig 1: how was the consensus determined? How do we know about the red arrows?

Protein similarity was predicted using the online software BLASTP by searching the Non-redundant protein sequences (nr) database of NCBI. The DNA binding domain of SrpA and the amino acids binding the DNA was displayed simultaneously by the program Conserved Domains in the same window. Conserved amino acids were predicted using an online tool WebLogo (<http://weblogo.berkeley.edu/>). One-thousand SrpA homologs with E value lower than 5E-08 were included in the assay. More details have been added in the sections of Materials and Methods and Results of the revised version.

12) Fig 2: Please indicate what is in lanes 1-4.

The necessary information has been added as follows.

Lane 1 (K5): phage K5 genomic DNA. Lane 2 (P/E): PAK-AR2 genomic DNA digested with *EcoRV*. Lane 3 (K5/E): K5 genomic DNA digested with *EcoRV*. Lane 4-7: total DNA samples from the PAK-AR2 cells infected with phage K5. Lane 8-11: total DNA samples from the 1-9 cells infected with phage K5. Lane 12-15: total DNA samples from the 1-9/pLLY1101 cells infected with phage K5.

13) Fig 4: This one does not show significant genes, only the selected pathways, or does it? It'd be probably better to present a bar chart of different pathways with the amount of up- and downregulated genes. The current figure is hard to interpret and colors are difficult to distinguish.

New figures 4b and 4c were generated to present bar charts as suggested by this reviewer.

14) Fig 9: I already commented on 9a. With 9b, I do not understand, why $E < 0.001$ was chosen for the BLAST search. This should result in many false positive homologs, since at this many low-level sequence similarities should show up that most likely do not represent homology.

The threshold was selected according to the description in the reference as shown below ^[1].

DNA:DNA alignment statistics are less accurate than protein:protein statistics; **while protein:protein alignments with expectation values < 0.001 can reliably be used to infer homology,** DNA:DNA expectation values $< 10^{-6}$ often occur by chance, and 10^{-10} is a more widely accepted threshold for homology based on DNA:DNA searches.

[1] Pearson, W. R. (2013). "An Introduction to Sequence Similarity ("Homology") Searching." Current protocols in bioinformatics / editorial board, Andreas D. Baxevanis ... [et al.] 0 3: 10.1002/0471250953.bi0471250301s0471250942.

15) Fig 10: This is very counterintuitive, since the colors indicate the effect of SrpA-deletion but the graph suggest the regulative effect of SrpA, which is the exact opposite. Some genes are shown as regulated that have no significantly differential expression in Table S3, e.g. qscR and lasI.

The color was indeed marked incorrectly, corrections have been made to the diagram in the revised version. The legend of figure 10 has been modified as “Figure 10. Schematic representation of the SrpA regulatory web in *P. aeruginosa*. The regulatory network includes the cellular processes and the genes significantly influenced by the SrpA regulator. It also includes the genes involved in the indicated cellular processes since they have the SrpA binding sites in their promoter regions. *PA1646* encodes a putative methyl-accepting chemotaxis transducer and highly expressed in the absence of SrpA. *PA3804* encodes a putative cytoskeleton protein RodZ required for the assembly of actin cytoskeleton MreB. Solid lines represent the downstream genes with the SrpA binding site in their promoters. Dot lines represent the direct or indirect regulation relationships that have been established previously. SrpA is depicted in oval textboxes. Regulators are depicted in diamond textboxes. Structural proteins or enzymes are depicted in elongated hexagon textboxes. Cellular processes are depicted in rectangle textboxes.”

Reviewer #2 (Remarks to the Author):

In this manuscript, the authors identify a bacterially-encoded protein (SrpA) that functions as a regulator of phage infection and a number of host processes including motility, interspecies interactions, and cell shape. The breadth but not depth of this

gene's distribution throughout the Pseudomonads and amongst other bacteria suggest horizontal transfer and perhaps particular function selected to increase resistance to phage while simultaneously increasing competitiveness against other microbial community members. This is an interesting story with strong supporting data. I have only a few comments/questions.

1) The authors base much of their reasoning, in terms of regulatory pathway alterations for their phenotypes, on work done in *P. aeruginosa* strains PAO1 and PA14. However, PAO1 and PA14 do not possess an ortholog of SrpA (personal search and the authors' Table S5). The authors should comment on this and how well they expect the SrpA-dependent regulation in strain PAK will translate to other *P. aeruginosa* strains. Optimally, the *srpA* gene should be expressed in PAO1 or PA14 and a few simple phenotypes examined (swarming and twitching) or, at very least, determine the conservation of the SrpA binding site in front of similar genes in one of these other strains.

That is a very good suggestion. The plasmid carrying the *srpA* gene was transferred into the PAO1 strain and the PA14 strain by electroporation. Cell motilities of the resulted strains were analyzed. The analysis showed that SrpA had no effects on swimming motility, inhibited swarming motility, and enhanced twitching motility in the tested PAO1 and PA14 strains, consistent with that observed in PAK-AR2 and its derivatives (Fig. S3).

2) Cell shape alterations are often pleiotropic and could explain the twitching motility, for instance, as longer cells tend to associate better and lead to more efficient

twitching. Perhaps I missed it, but this should be a caveat to be included.

We agree that cell shape is tightly linked with cell motilities. During swimming process, bacterial cells coordinately differentiate into elongated and hyperflagellated swarmer cells which are required for the subsequent swarming motility ^[1,2]. For twitching motility, in *Legionella pneumophila*, the deletion of the *fimV* gene resulted in loss of twitching motility and cell elongation ^[3], while in *P. aeruginosa*, the *fimV* gene was also confirmed to be required for type IV pili (Tfp) mediated twitching motility ^[4]. However, to our best knowledge, the relationship between the *fimV* gene and cell shape has not been investigated. Only one paper described that 'overexpression of *fimV* results in the formation of dramatically elongated cells' which are 'only a small percentage of the population demonstrated this abnormality, and cells at the center of the colony, away from the twitching edge, appeared largely normal'. No scanning electron microscope (SEM) images were shown in the paper ^[5].

In our work, cell shape alterations were one of the multiple phenotypic changes observed in the *srpA* gene disruption mutant 1-9. More experiments are required to investigate the connections among these phenotypic changes. Accordingly, no elaboration on these have been included in the discussion section.

[1] Harshey, R. M. (1994). "Bees aren't the only ones: swarming in Gram-negative bacteria." *Molecular microbiology* 13(3): 389-394.

[2] Merino, S., et al. (2006). "Bacterial lateral flagella: an inducible flagella system." *FEMS Microbiol Lett* 263(2): 127-135.

[3] Coil, D. A. and J. Anne (2010). "The role of *fimV* and the importance of its

tandem repeat copy number in twitching motility, pigment production, and morphology in *Legionella pneumophila*." *Arch Microbiol* 192(8): 625-631.

[4] Wehbi, H., et al. (2011). "The peptidoglycan-binding protein FimV promotes assembly of the *Pseudomonas aeruginosa* type IV pilus secretin." *J Bacteriol* 193(2): 540-550.

[5] Semmler, A. B., et al. (2000). "Identification of a novel gene, *fimV*, involved in twitching motility in *Pseudomonas aeruginosa*." *Microbiology* 146 (Pt 6): 1321-1332.

3) Fig3d: For the EMSAs, where is the rest of the band shifting as SrpA is added for Pgp058? Presumably to higher order structures. It would be useful to show the entire EMSA image for just this promoter region.

The gel-shifting images have been replaced with the full EMSA images. The molar concentration of SrpA and the corresponding DNA fragments labeled with the fluorescent agent cy3 were marked for each binding reaction. The changes have been made in the sections of Material and Methods, Results, and figure legends. And poly (dI-dC) was used in the reactions to reduce non-specific bindings (Fig. 2d and S2).

4) Writing quality: The Results section is reasonable and the Discussion section borderline, but the Introduction has a number of grammatical issues and needs heavy copy editing.

The revised version has been thoroughly edited by one of our colleagues who is a native English speaker.

Reviewer #3 (Remarks to the Author):

You et al. NCOMMS-17-24307

Summary This manuscript describes the identification and subsequent analyses of SrpA, which controls both phage infectivity and core cellular processes in *Pseudomonas aeruginosa*. Overall, this is an interesting study but the authors should consider the following issues:

Review criteria

a. What are the major claims of the paper?

- The authors identified a bacterial encoded protein, SrpA, which can regulate both phage and bacterial genes.
- It is claimed that SrpA can monitor core cellular processes in response to phage infection and environmental signals, providing insights into other such proteins that are widespread in bacteria without any known function.

b. Are the claims novel and will they be of interest to others in the community and the wider field?

- Yes I feel the information conveyed in Fig. 9 could be of broad interest to the community and wider field. It is quite interesting that these proteins are widespread.
- I feel the paper would have been more impactful if the authors could better tie in the biology of the phage with the list of bacterial genes that are co-regulated by SrpA.

Why does SrpA regulate this subset of bacterial genes and how is this important for the phage life cycle?

It is a great question and very hard to interpret based on the available data. SrpA can bind directly to the promoter of the phage RNA polymerase gene and recruit the RNA polymerase to initiate transcription of this gene and other phage genes. Without SrpA, phage proliferation becomes too slowly to produce novel progenies. On the other hand, SrpA controls a series of core cellular processes including T3SS, T6SS, pyocyanin synthesis, biofilm formation, swarming, twitching, worm killing efficiency, and cell shapes by direct or indirect regulations. The alterations of the core cellular processes will bring about different competitiveness under certain environmental stresses. The phylogenetic analysis showed that SrpA is widely distributed in the world of microbes, especially in bacteria, suggesting the *srpA* gene may originate from bacteria. One common view about the origin of phages is that they are originated from bacteria^[1,2]. Phage K5 and its possible ancestors may be from *Pseudomonas aeruginosa*. A short discussion of this part has been added in the revised manuscript.

[1] Brüssow, H., et al. (2004). "Phages and the Evolution of Bacterial Pathogens: from Genomic Rearrangements to Lysogenic Conversion." *Microbiology and Molecular Biology Reviews* 68(3): 560-602.

[2] Hendrix, R. W., et al. (2003). "Bacteriophages with tails: chasing their origins and evolution." *Research in Microbiology* 154(4): 253-257.

c. Is the work convincing?

- The manuscript is a bit challenging to read and could use English language editing.

Significant editorial work throughout seems necessary.

The revised version has been edited by a native English speaker.

- Fig. 2b, label the top of the gel as time of DNA harvest (minutes). What is in lane 1 and 3?

Appropriate labels were added for lane 1 to lane 3.

- Data in Fig. 3a is redundant with what is shown in Fig. 2 d-f. Why are both necessary? Also, the fusions being used to selected genes in Fig 3b are not labeled. Are they same as those being evaluated in Fig. 3c?

Fig. 3a was replaced with a new figure showing one unknown viral factor may improve the expression of the phage RNA polymerase gene. Changes have been made in the sections of Material and Methods, Results, and the figure legend. The transcriptional fusions used in Fig. 3b were the same as that in Fig. 3c and they have been added in the Fig. 3b.

- I don't feel that the data described in Fig. 3 and Fig. S1 justify the statement that SrpA and the phage RNA polymerase possibly co-regulate transcription of phage genes (line 350 and title on line 338). If SrpA only controls expression of the phage polymerase then the effects on these other genes would be indirect. Indirect effects also seem to be supported by the EMSA experiment in Fig. 3d that shows SrpA only binds to gp058 promoter DNA. The key experiment would be to uncouple phage RNA

polymerase expression from SrpA and then evaluate whether these phage genes still require SrpA for expression.

The plasmid pJJY1601, carrying the phage RNA polymerase gene *gp058* with its own native promoter which contains the binding site for SrpA, was constructed. In PAK-AR2 cells, presence of the SrpA enhanced the transcription of the *gp058* gene by recruiting the phage RNA polymerase. With the increased expression of the phage RNA polymerase, the transcription level of all phage gene transcription fusions was increased as shown in figure 3b. While in 1-9 cells, SrpA is absent and the transcription level of the *gp058* gene is at a relatively low level, resulting in low amount of the phage RNA polymerase for the transcription of all phage genes including *gp058* as shown in figure 3c.

As suggested, the plasmid pYXJ1701, carrying the *gp058* gene driven by the P_{lac} promoter which does not contain the SrpA binding site, was constructed. Expression of two phage genes was evaluated, one was $P_{gp058-lacZ}$ which contains a SrpA binding site and the other was $P_{gp105-lacZ}$ which lacks a SrpA binding site (Fig. 3f). In 1-9 cells, the plasmid pYXJ1701 produced enough phage RNA polymerase, while the plasmid pYJJ1605 not since it needs SrpA for full expression. The results showed that the expression of *gp058* requires the presence of SrpA, while that of *gp105* does not.

- Fig. 3e, are the numbers shown relative to the start of *gp058* transcription?

The numbers are the relative distances upstream to the first start codon ATG of the *gp058* coding region, not the transcription initiation site.

- Fig. 4. I might suggest using an additional color to represent those phage K5 genes that were also found to be regulated in these two strains. It would appear these are all host (bacterial genes).

The total RNA samples for the transcriptome analysis were extracted from the cells of PAK-AR2 and 1-9, respectively. Phage K5 was not added in this experiment and thus no phage genes were detected in the analysis.

To give more details of the transcriptome assay, two additional figures 4b and 4c were generated to illustrate the expression profiles of the genes classified according to the KEGG_B_class (Fig. 4b) and the genes of the major pathways (Fig. 4c).

- Fig. 6. I found it strange that the impact of purEK mutation present in strain PAK-AR2 was only investigated for pyocyanin production and virulence. What is the rationale for this and what is the explanation for this phenotype? This seems dropped in the manuscript and the AR2/1-9 vs. PAK/C11-1 comparisons are only selectively used throughout the paper. Are the cell morphology differences described in Fig. 7d due to SprA or purEK? The length differences are not fully restored upon complementation.

Plasmid conjugation was used to construct the Tn5G insertion library of the PAK strain. But *E. coli* cells (donor) contamination happens occasionally during the selection of the phage resistant mutants. PAK-AR2, a derivative of PAK, harbors a Ω fragment which confers spectinomycin/streptomycin resistance and was used as the recipient strain for construction the Tn5G insertion bank. And 1-9 was selected from

this bank.

Most experiments in our study showed pronounced effects of SrpA among the strain PAK-AR2 and 1-9 with a few exceptions, including pyocyanin synthesis and worm killing assay. The *purEK* deletion seems overlapping the effects of the *srpA* disruption in those tests. To test this possibility, the strain PAK and C11-1 were included in the assays. And the results show that SrpA does regulate these metabolic pathways.

We agree that the length differences are not fully restored upon complementation. By reviewing the data, we found some cells were in division stage and their length are generally shorter than individual cells. The data was re-analyzed by removing the dividing cells from the data sets. The figure 7d was replaced with newer version. No significant length difference was obtained between the cell groups of PAK-AR2 and 1-9/pLLY1101 ($P > 0.05$), suggesting the *srpA* gene can fully restore the length differences by complementation.

- Fig. S2, these data are not convincing without appropriate controls including competing DNA molecules to show specificity. Why do the DNA binding activities of SrpA change so much for gp053 and gp125 at 40 ng/ul? What is the SrpA molar concentration used here?

As suggested, gel-shifting assay was repeated by including the poly (dI-dC) to reduce non-specific binding. To increase the sensitivity of the binding assay, the fluorescence-labeled DNA fragments were used in the assay. Figure 2d and S2 have been replaced with new images. The molar concentrations of SrpA and the

corresponding DNA fragments have been marked for each binding reaction.

d. Appropriateness and validity of the statistics.

- Most seem appropriate but I don't understand how there is a statistically significant difference in length between the strains being evaluated in panel 7d. The standard errors seem to overlap yet the p value is <0.001 .

ANOVA (analysis of variance) was used for statistics analysis. Since each sample group includes 40 cells for analysis, the more samples included and the more confidential of the data. If less samples were used in the analysis, p value would increase dramatically.

Reviewers' comments:

Reviewer #1 (Remarks to the Author):

The authors have done a good job in improving their manuscript, which helps a lot to understand the results. The interconnection of SrpA with the regulatory network of *P. aeruginosa* is evident but nevertheless surprising, given that the gene itself is only present in a fraction of the known *P. aeruginosa* strains while the binding site seems to be ubiquitous. The added presentation of the genomic context (Fig. S3) in PAK in comparison with PAO1 and PA14 emphasizes the putative origin of SrpA by horizontal gene transfer. This supports a hypothetical coevolution of SrpA with *P. aeruginosa* specific phages. A possible task for future work will be to look for correlations of phage specificity with the presence of SrpA in the different host genomes.

The implications of the presented results are highly interesting and will have impact in the *Pseudomonas* community and likely beyond, considering the evolutionary aspects of the story. Therefore, I recommend the manuscript for publication but there are still some points that could be improved:

1) The additional clean deletion of the *srpA* gene and repeating the experiments done with the original mutant 9-1 shows that the observed effects were no artifacts. If the authors choose to keep this in the supplementary section, they should at least add some reference to it in the text. I only found it mentioned in the caption of Fig S5 and in the Acknowledgments. Also, the deletion methods are not described. Mentioning them in the supplementary section would be sufficient but it should not be omitted.

2) The statistics are still not sufficiently described. If ANOVA was used, why are multiple pairwise comparisons presented, e.g., in figure 5? Are these the results of post-hoc tests or was the T-test used here? If so, did the method include a correction for multiple testing?

3) Figure 1: I still don't see the origin of the red arrows. How do the authors know that these Amino Acids recognize specific DNA sequences? Is this a known feature of the HTH motif?

4) Figure 3 and S2: If I calculated correctly, the concentrations chosen for the EMSA of Pgp058 are now in a lower range than in the original version. I find it most interesting that the shift now seems to gradually increase, an observation that also occurs for some but not all of the other oligos that were tested. What is the explanation for this behaviour? Does this hint towards multiple proteins binding to each promoter or is there a concentration dependent change in the structure of the DNA:Protein complex?

5) Figure 9: A very important figure but there are some problems:

- a. The taxonomic levels presented in different colors are used inconsistently. Some represent qualified taxonomic names (gamma-proteobacteria) and some not (Enterobacteria), and many of them represent different taxonomic levels that may even overlap (Enterobacteriaceae > Gamma-Proteobacteria > Proteobacteria > Bacteria). Thus, the meaning of the different colors is not clear. The yellow box in front of SrpA should be replaced by the red arrow that indicates its location in the tree.
- b. The legend should describe the three different data sets. It makes no sense to write *P. aeruginosa* two times.
- c. This is a very interesting figure that is supposed to show the sketchy distribution of SrpA within the species *P. aeruginosa*. However, it is not suitable to make a phylogenetic tree of strains of a single species based on the 16S rRNA gene. There is no sufficient variation to establish a meaningful phylogenetic relation of so many closely related strains. If the tree were displayed with branch lengths

presenting actual phylogenetic distances, the tree would be very flat, I suppose. I'd strongly suggest to replace this tree by one based on another gene, or better multiple genes. Suitable alternatives to test include other essential genes with a good tradeoff between conservation and variation, e.g., *atpD*, *rpoB*, or the genes used in established MLST methods for *P. aeruginosa* (see Curran, B., et al., (2004). *J Clin Microbiol* 42(12): 5644-5649; and Vernez, I., et al., (2005). *FEMS Immunol Med Microbiol* 43(1): 29-35.). It may be difficult to retrieve as many sequences from the databases as in the case of the 16S rRNA but the tree will have a much better resolution and will be more meaningful and robust.

Reviewer #2 (Remarks to the Author):

The authors have responded appropriately to all of my questions from the first review.

Reviewer #3 (Remarks to the Author):

I feel that the authors have adequately addressed the critiques and questions raised in the prior review including my concerns. The presentation and grammatical issues have been largely rectified.

Responses to Referees' Comments

Reviewer #1 (Remarks to the Author):

1) The additional clean deletion of the *srpA* gene and repeating the experiments done with the original mutant 9-1 shows that the observed effects were no artifacts. If the authors choose to keep this in the supplementary section, they should at least add some reference to it in the text. I only found it mentioned in the caption of Fig S5 and in the Acknowledgments. Also, the deletion methods are not described. Mentioning them in the supplementary section would be sufficient but it should not be omitted.

The deletion method for construction of the mutant strain KO4 was described in the footnote of Table S1. The result about the clean deletion mutant KO4 has been described in line 9 on Page 27, as “To further confirm the function of the *srpA* gene, a mutant strain completely deleted of the *srpA* gene (KO4) was generated (Table S1). In cell motility assay, KO4 displayed similar phenotypes as the strain 1-9 (Fig. S5), indicating that the *srpA* gene was fully inactivated in both strains.”

2) The statistics are still not sufficiently described. If ANOVA was used, why are multiple pairwise comparisons presented, e.g., in figure 5? Are these the results of post-hoc tests or was the T-test used here? If so, did the method include a correction for multiple testing?

All of the statistical analysis was conducted by one-way ANOVA, including the data shown in the Figure 5, there were no multiple pairwise comparisons. Corrections have

been made to replace “ANOVA” with “one-way ANOVA” throughout the manuscript.

3) Figure 1: I still don't see the origin of the red arrows. How do the authors know that these Amino Acids recognize specific DNA sequences? Is this a known feature of the HTH motif?

Sequence-specific DNA binding sites are predicted by the BLASTP software upon search of the NCBI database, as the green arrows shown in the following image.

The green arrows displayed in the above image were replaced with red arrows in the Figure 1. The location of the red arrows is now precisely positioned in the new version of the Figure 1.

4) Figure 3 and S2: If I calculated correctly, the concentrations chosen for the EMSA of Pgp058 are now in a lower range than in the original version. I find it most interesting that the shift now seems to gradually increase, an observation that also occurs for some but not all of the other oligos that were tested. What is the explanation for this behavior? Does this hint towards multiple proteins binding to each promoter or is there a concentration dependent change in the structure of the DNA:Protein complex?

The concentrations of SrpA used in the new experiments were the same as the previous experiments, corrections have been made for the Figure 3d and S2. As to the

dose dependent gradual increase in shifting, it might be due to the conformational change of the DNA-protein complex and/or the number of SrpA binding to the target fragment. A short discussion has been added on this topic on Page 24.

5) Figure 9: A very important figure but there are some problems:

a. The taxonomic levels presented in different colors are used inconsistently. Some represent qualified taxonomic names (gamma-proteobacteria) and some not (Enterobacteria), and many of them represent different taxonomic levels that may even overlap (Enterobacteriaceae > Gamma-Proteobacteria > Proteobacteria > Bacteria). Thus, the meaning of the different colors is not clear. The yellow box in front of SrpA should be replaced by the red arrow that indicates its location in the tree.

With the Algorithm PSI-BLAST (Position-Specific Iterated BLAST), the phylogenetic tree of Figure 9a was generated based on the sequence of the SrpA-like proteins, and it was displayed in the window of Blast Tree View. The clades displayed in the phylogenetic tree include homologs from highly diverse microorganisms (Table S5), and some groups cannot be assigned a qualified taxonomic name. The clades shown in this figure were labeled with 'blast names and colors' provided by the program PSI-BLAST as follows.

high GC Gram+
planctomycetes
b-proteobacteria
g-proteobacteria
bacteria
cyanobacteria
green sulfur bacteria
d-proteobacteria
verrucomicrobia
CFB group bacteria
proteobacteria
a-proteobacteria
archaea
enterobacteria
unknown

The yellow box in front of SrpA was replaced with red triangle as suggested.

b. The legend should describe the three different data sets. It makes no sense to write *P. aeruginosa* two times.

The legend has been modified as suggested.

c. This is a very interesting figure that is supposed to show the sketchy distribution of SrpA within the species *P. aeruginosa*. However, it is not suitable to make a phylogenetic tree of strains of a single species based on the 16S rRNA gene. There is no sufficient variation to establish a meaningful phylogenetic relation of so many closely related strains. If the tree were displayed with branch lengths presenting actual phylogenetic distances, the tree would be very flat, I suppose. I'd strongly suggest to replace this tree by one based on another gene, or better multiple genes. Suitable alternatives to test include other essential genes with a good tradeoff between conservation and variation, e.g., *atpD*, *rpoB*, or the genes used in established MLST methods for *P. aeruginosa* (see Curran, B., et al., (2004). *J Clin Microbiol* 42(12): 5644-5649; and Vernez, I., et al., (2005). *FEMS Immunol Med Microbiol* 43(1):

29-35.). It may be difficult to retrieve as many sequences from the databases as in the case of the 16S rRNA but the tree will have a much better resolution and will be more meaningful and robust.

As suggested, the sequences of the genes *atpD* (1377 bp), *rpoA* (1002 bp), *recA* (1041 bp), *recN* (1677 bp), *pilA* (450 bp), *oprD* (1332 bp), *oprN* (1419 bp), and *mexE* (1245 bp) were retrieved from the *Pseudomonas* genome database. However, the BLAST results showed that all these genes are as highly homologous as the 16S rRNA gene among those ~2000 strains. The phylogenetic trees of the genes *atpD*, *pilA*, *oprD*, *oprN*, and *mexE* were also constructed, and they showed a flat pattern similar to that of the 16S rRNA gene. We failed to retrieve the sequences of other genes, such as *rpoB*, *rpoC*, and *rpoD*, due to the size of these genes, too large for conventional desktop computers to run. As reported previously, “*P. aeruginosa* is endowed with a highly conserved core genome of low sequence diversity and a highly variable accessory genome that communicates with other pseudomonads and genera via horizontal gene transfer” ^[1], it is extremely difficult to find suitable target genes for discrimination of the *P. aeruginosa* strains. Therefore, we prefer to keep the original phylogenetic tree based on the 16S rRNA gene.

[1] Klockgether J, Cramer N, Wiehlmann L, Davenport CF, Tummler B (2011) *Pseudomonas aeruginosa* Genomic Structure and Diversity. *Frontiers in microbiology* 2:150.